# Identifying healthy individuals with Alzheimer's disease neuroimaging phenotypes in the UK Biobank

Tiago Azevedo [1], Richard A. I. Bethlehem [2,3], David J. Whiteside[4], Nol Swaddiwudhipong [4], James B. Rowe[4], Pietro Lió [1], Timothy Rittman [4✉] & the Alzheimer's Disease Neuroimaging Initiative*

## Abstract

**Background** Identifying prediagnostic neurodegenerative disease is a critical issue in neurodegenerative disease research, and Alzheimer's disease (AD) in particular, to identify populations suitable for preventive and early disease-modifying trials. Evidence from genetic and other studies suggests the neurodegeneration of Alzheimer's disease measured by brain atrophy starts many years before diagnosis, but it is unclear whether these changes can be used to reliably detect prediagnostic sporadic disease.

**Methods** We trained a Bayesian machine learning neural network model to generate a neuroimaging phenotype and AD score representing the probability of AD using structural MRI data in the Alzheimer's Disease Neuroimaging Initiative (ADNI) Cohort (cut-off 0.5, AUC 0.92, PPV 0.90, NPV 0.93). We go on to validate the model in an independent real-world dataset of the National Alzheimer's Coordinating Centre (AUC 0.74, PPV 0.65, NPV 0.80) and demonstrate the correlation of the AD-score with cognitive scores in those with an AD-score above 0.5. We then apply the model to a healthy population in the UK Biobank study to identify a cohort at risk for Alzheimer's disease.

**Results** We show that the cohort with a neuroimaging Alzheimer's phenotype has a cognitive profile in keeping with Alzheimer's disease, with strong evidence for poorer fluid intelligence, and some evidence of poorer numeric memory, reaction time, working memory, and prospective memory. We found some evidence in the AD-score positive cohort for modifiable risk factors of hypertension and smoking.

**Conclusions** This approach demonstrates the feasibility of using AI methods to identify a potentially prediagnostic population at high risk for developing sporadic Alzheimer's disease.

## Plain language summary

Spotting people with dementia early is challenging, but important to identify people for trials of treatment and prevention. We used brain scans of people with Alzheimer's disease, the commonest type of dementia, and applied an artificial intelligence method to spot people with Alzheimer's disease. We used this to find people in the Healthy UK Biobank study who might have early Alzheimer's disease. The people we found had subtle changes in their memory and thinking to suggest they may have early disease, and we also found they had high blood pressure and smoked for longer. We have demonstrated an approach that could be used to select people at high risk of future dementia for clinical trials.

[1] Department of Computer Science and Technology, University of Cambridge, Cambridge, UK. [2] Brain Mapping Unit, Department of Psychiatry, University of Cambridge, Cambridge, UK. [3] Autism Research Centre, Department of Psychiatry, University of Cambridge, Cambridge, UK. [4] Department of Clinical Neurosciences and Cambridge University Hospitals NHS Trust, University of Cambridge, Cambridge, UK. *A list of authors and their affiliations appears at the end of the paper. ✉email: tr332@medschl.cam.ac.uk

A critical task in dementia research is to identify disease at the earliest possible time point, permitting early intervention with lifestyle change[1] or disease-modifying therapies[2] at a time when the disease process could potentially be reversed or halted, and quality of life remains high. The difficulty in achieving early and accurate diagnosis has been highlighted as a major factor in the lack of success of clinical trials for neurodegenerative diseases, including Alzheimer's disease[2,3]. Neuroimaging abnormalities in genetic dementia cohorts suggest that neurodegenerative pathologies begin decades before symptoms[4,5]. Predicting disease with such certainty before symptom onset is not possible in sporadic forms of dementia, so an alternative strategy is needed to identify an at-risk population using disease biomarkers to find people with early stages of neuropathology who are at high risk of developing cognitive impairment in the future. Small studies of ageing cohorts capturing people who have converted to Alzheimer's disease have identified group-level structural changes in the medial temporal lobe detectable prior to diagnosis[6–9]. Whether identifying such changes are sufficient to identify individuals at risk of dementia is unclear. Identifying such a high-risk group would be suitable for prevention studies or early disease-modifying treatment trials[10].

The challenge of identifying disease at the earliest possible point has led to proposed criteria for at-risk or presymptomatic Alzheimer's disease that rely on biomarker evidence rather than clinical syndrome[11]. One set of criteria proposes an "ATN" classification of Alzheimer's disease, representing Amyloid (A), Tau (T), and Neuronal loss (N) as central pillars of Alzheimer's pathology[12]. Many biomarkers to assess brain tau and amyloid pathology in life are expensive, invasive, or not widely available. However, neuronal loss is readily measured in vivo using structural brain imaging.

Structural neuroimaging has been central in clinical diagnosis and in attempts to classify Alzheimer's disease using neuroimaging for many years[13,14]. Loss of volume in the hippocampus is well described in Alzheimer's disease, and whole brain volume may also be relevant[15,16]. Other specific brain regions are less well studied, yet may be relevant in identifying people with Alzheimer's disease—alone or combined with hippocampal atrophy. More complex analytical approaches offer the opportunity to use all the available information from structural neuroimaging data to identify a specific pattern of atrophy relevant to the disease.

Artificial Intelligence (AI) and Machine learning (ML) describe computational algorithms that can make predictions that reflect intuitive human thinking and can 'learn' from new data. A class of AI models called deep learning methods uses multiple hierarchical levels of data abstraction to identify important features to make a prediction. These models have been successfully applied to several different contexts in medicine[17,18], leveraging neuroimaging datasets[19] to address a multitude of neuroscientific questions[20]. AI approaches in structural MRI have facilitated the classification of Alzheimer's disease with a good degree of accuracy[21–30], but few such studies have validated their approach in an independent dataset[31,32].

Despite these achievements in the neuroimaging field, there are challenges to the generalisability of deep learning models[33,34]. A relatively recent trend applies a probabilistic approach to deep learning by using measures to describe Bayesian uncertainty[35,36]. Such uncertainty measures allow for a better characterisation of the model's output rather than solely a deterministic value[37], ultimately strengthening the confidence in results derived from these stochastic models[38].

Probabilistic AI approaches are strong candidates to make the best use of all available information in structural neuroimaging. The availability of large open-access neuroimaging repositories permits us to use distinct datasets for training an AI model and assessing its generalisability. In this work, we use a selective and well-characterised dataset of Alzheimer's disease to train the model (Alzheimer's Disease Neuroimaging Initiative dataset, ADNI), and a more 'noisy' real-world clinical dataset with a range of different diseases to assess generalisability (National Alzheimer's Coordinating Center, NACC).

To identify a group of people at high risk of developing dementia, we use the trained model to find people with a neuroimaging AI-derived phenotype of Alzheimer's disease in a healthy cohort without a diagnosis of dementia from the UK Biobank study. We demonstrate poorer cognitive performance in people with an AD-like neuroimaging profile, consistent with a high prevalence of early Alzheimer's pathology and potentially suitable for screening and selection into disease-modifying trials. We find that this group reports poorer general health and identify hypertension and smoking as modifiable risk factors in this cohort.

## Methods

**Datasets**. The ADNI study recruits people with Alzheimer's disease, Mild Cognitive Impairment (MCI), and control participants. It is primarily a research cohort and has a well-characterised population who have undergone high-quality, standardised neuroimaging with a standard battery of cognitive and clinical assessments. We used all available datasets from ADNI1, ADNI2, ADNI-GO, and ADNI3, comprising 736 baseline scan sessions from the ADNI dataset with a diagnosis of Alzheimer's disease ($n = 331$) and Controls ($n = 405$), whose demographic data are summarised in Table 1.

Because ADNI is a relatively select research cohort, it is vulnerable to selection bias[39]; therefore, we used the NACC dataset for validation. This NACC dataset is a "real-world" memory clinic-based cohort, including people with Alzheimer's disease and a range of other cognitive and non-cognitive disorders. Because of its pragmatic nature, the NACC dataset is more heterogeneous in the quality of imaging and additional data collected. Therefore, it is an ideal dataset for validation of a tool developed in a more 'clean' dataset such as ADNI. We used 5209 people from the NACC dataset whose demographics are summarised in Table 2. Other degenerative disorders (OD) correspond to NACC labels "Vascular brain injury or vascular dementia including stroke", "Lewy body disease (LBD)", "Prion disease (CJD, other)", "FTLD, other", "Corticobasal degeneration (CBD)", "Progressive supranuclear palsy (PSP)", and "FTLD with motor neuron disease (e.g., ALS)". Other non-degenerative

**Table 1 Summary of demographics of the ADNI dataset.**

| Diagnosis | *n* | Mean age (s.d.) | Sex (male/female) |
|---|---|---|---|
| AD | 331 | 75 (7.8) | 181/150 |
| Control | 405 | 74.7 (5.7) | 202/203 |

**Table 2 Summary of demographics of the NACC dataset.**

| Diagnosis | *n* | Mean age (s.d.) | Sex (male/female) |
|---|---|---|---|
| Control | 2824 | 68.6 (10.9) | 938/1886 |
| AD | 1706 | 73.9 (9) | 794/912 |
| Other degenerative disorders | 326 | 71.2 (9.9) | 196/130 |
| Other non-degenerative disorders | 353 | 69.1 (10) | 135/218 |

**Table 3 Demographics for those in the UK Biobank who underwent neuroimaging.**

| n | Mean age (s.d.) | Sex (male/female) |
|---|---|---|
| 37,104 | 55.3 (7.4) | 19,493/17,611 |

disorders (OND) correspond to NACC labels "Depression", "Other neurologic, genetic, or infectious condition", "Cognitive impairment for other specified reasons (i.e., written-in values)", "Anxiety disorder", "Cognitive impairment due to medications", "Other psychiatric disease", "Cognitive impairment due to systemic disease or medical illness", "Traumatic brain injury (TBI)", "Cognitive impairment due to alcohol abuse", "Bipolar disorder", and "Schizophrenia or other psychosis".

We deliberately chose not to train the model on the NACC dataset. The argument to train on this dataset rather than ADNI would be the larger dataset available, but the NACC dataset is much more 'noisy' in the sense that the diagnostic labels are clinical rather than biomarker supported (as in ADNI). It is highly likely that if we had trained in the NACC data and validated in the ADNI dataset our results would have looked better in terms of raw accuracy, but we consider that this would be falsely reassuring given the highly selected nature of the ADNI cohort.

Finally, we used the UK Biobank as a non-clinical longitudinal ageing cohort to apply the algorithm to a healthy cohort. This dataset is subject to potential selection bias, tending to be a population with a low risk for disease[40]. Despite these limitations, the size of the dataset, the age of participants, and the high-quality neuroimaging data make it an ideal cohort to assess at-risk features for neurodegenerative disease. A summary of the UK Biobank neuroimaging data is found in Table 3. Some cognitive tests used in the UK Biobank are not clinical-standard tests but include fluid intelligence (Touch-screen fluid intelligence test. https://biobank.ctsu.ox.ac.uk/crystal/ukb/docs/Fluidintelligence.pdf), numeric memory (Touch-screen numeric memory test. https://biobank.ctsu.ox.ac.uk/crystal/ukb/docs/numeric_memory.pdf), matrix reasoning (UK Biobank Category 501—matrix pattern completion. https://biobank.ctsu.ox.ac.uk/crystal/label.cgi?id=501), and reaction time (UK Biobank Category 100032—reaction time. https://biobank.ctsu.ox.ac.uk/crystal/label.cgi?id=100032). The validity of these tests has been assessed separately, finding moderate to high validity for the cognitive tests used[41].

The previous three datasets are provided by the respective consortium and not collected by us; therefore, each consortium had its relevant ethical regulations obtained and approved, and no extra ethical regulations were specifically required for this work.

**ADNI preprocessing**. In each cohort, structural MRI Magnetization Prepared—RApid Gradient Echo (MPRAGE) scans were acquired. Further details of the individual imaging protocols are available for ADNI at http://adni.loni.usc.edu/methods/documents/mri-protocols/, for NACC at https://files.alz.washington.edu/documentation/rdd-imaging.pdf, and for the UK Biobank at[42]. Scans underwent estimation of regional cortical volume, regional cortical thickness, and estimated total intracranial volume using the FreeSurfer toolbox (version 6.0)[43]. Given the size of the cohorts, the resulting segmentations were assessed for gross abnormalities, but minor registration errors were not corrected. Results were obtained for cortical thickness and volume in the 68 surface-based regions of the Desikan–Killiany atlas from both hemispheres. In addition, the brainstem volume was also extracted together with 9 volume features per hemisphere (cerebellum white matter, cerebellum cortex, thalamus proper, caudate, putamen, pallidum, hippocampus, amygdala, and accumbens area). In total, 155 features were extracted per brain scan.

The ADNI dataset was divided into a training set with 662 samples and a validation set with 74 samples, representing approximately 90% and 10% of the original cohort, respectively. This division approximately preserved the relative distributions of diagnosis, estimated total intracranial volume, sex, and age.

To regress out confounds, 155 distinct linear regression models (one for each input feature) were fitted to the training set using ordinary least squares (OLS) implemented in *statsmodels*[44]. For each of the 68 cortical thickness features, the independent variable to be regressed out was age. For the remaining 87 volume features, the independent variables were age, estimated total intracranial volume, and sex. These 155 regression models (as defined using the *statsmodels* package) were saved in disk to be later employed on the ADNI validation set, NACC, and UK Biobank datasets. We ensure no data leakage in the training and evaluation processes by deconfounding the validation/test sets (i.e., NACC, UK Biobank, and ADNI validation set) using only ADNI confounding factors learned from the training set.

After the data is deconfounded, each 155 input feature was separately scaled to zero mean and unit variance for numerical stability when training a neural network using *Scikit-learn*[45]. Normalisation statistics were once again calculated only using the ADNI training set. Values in the validation/test sets (i.e., NACC, UK Biobank, and ADNI validation set) were normalised using the normalisation statistics from the ADNI training set, after the deconfound process.

**Bayesian machine learning**. A supervised machine learning (ML) model learns a target function $f_\theta$, parameterised by $\theta$, such that it can predict $y = f_\theta(x)$. In the case of a classification task, the function is such that $f : \mathbb{R}^N \to \{1, \dots, k\}$, where $k$ is the number of possible categories (i.e. labels). For example, for a certain image with pixels represented in a feature vector $x$, the function could try to predict whether it contains a dog, a cat, or a bird ($k = 3$); in our context, the binary classification model predicts whether a patient has Alzheimer's disease or not ($k = 2$). Practically, this function $f_\theta$ learns how to predict labels $y$ from features $x$ by estimating the probability distribution $p(y|x)$ that generated those same labels.

The function $f_\theta$ can be modelled as a deep neural network. To train such a model with a particular dataset, one needs to tune the learnable parameters of that model (i.e. $\theta$) by minimising a loss function using stochastic gradient descent or another optimisation algorithm. In contrast, under Bayesian ML the Bayes rule is used to infer model parameters $\theta$ from data $x$:

$$p(\theta|x) = \frac{p(x|\theta)p(\theta)}{p(x)}. \tag{1}$$

Here, the model parameters are represented by the posterior distribution $p(\theta|x)$, where the model parameters $\theta$ are conditioned on the data $x$. The goal of Bayesian ML is then to estimate this distribution given the likelihood $p(x|\theta)$ and the prior distribution $p(\theta)$ (i.e. belief of what the model parameters might be). The prior $p(x)$ cannot be generally computed but as it is a normalising constant not dependent on $\theta$ and it stays the same for any model, it can be dropped from calculations when estimating the posterior. The posterior distribution cannot usually be analytically calculated using big data in a practical way, and therefore there are several methods to calculate these distributions and approximate the intractable posterior[46].

We use Monte Carlo dropout[47,48] to approximate Bayesian inference by using dropout during the inference phase of model[49]. Dropout is a regularisation approach often employed

**Fig. 1 Architecture of the neural network model used in this paper.** The neural network consists of two hidden layers of 128 dimensions and non-linear activation function $\sigma = \tanh()$. For each set of 155 inputs, $N = 50$ forward passes are run, each time with a different dropout mask sampled from a Bernoulli distribution. An Alzheimer's disease likelihood score is generated as the mean, and model uncertainty as the standard deviation calculated from the 50 forward passes.

in deep neural networks to avoid overfitting which works by randomly dropping nodes during the training process. With Monte Carlo dropout, nodes are also randomly dropped during inference which means that for the same input, each forward pass will generate a different output; this is possible as for each pass a different Bernoulli mask is applied to the neural network's weights. Gal and Ghahramani[47] show that each forward pass on the neural network corresponding to a different dropout mask is a good approximation to sampling from the true posterior distribution $p(\boldsymbol{\theta}|\boldsymbol{x})$.

With this simple yet powerful approximation, one can have the statistical power of a Bayesian ML model at very little added computational cost. Indeed, the Monte Carlo dropout method was chosen as it works well on a wide variety of previously trained neural networks; therefore, it could be used in other clinical contexts without the requirement for a complete knowledge of Bayesian statistics. Furthermore, Monte Carlo dropout is known to bring advantages in modelling uncertainty[47], which is of paramount importance in a clinical context, as well as better overall performance for certain downstream tasks[50].

**Deep neural network implementation**. As depicted in Fig. 1, we implemented a neural network with two hidden layers, each with 128 dimensions. Given the small dataset size, we empirically found these hyperparameters to give stable learning curves, thus avoiding a deeper neural network that could more easily overfit the small training data. Dropout layers were added after each hidden layer with a high dropout rate (i.e., 80%) to help in avoiding overfitting given the small neural network size; a smaller dropout rate was empirically found to provide slightly worse metrics on the ADNI validation set. We used the hyperbolic tangent function (tanh()) as the non-linear activation function to leverage both the positive and negative value ranges of the input. This non-linear activation is a symmetric function in which negative inputs will be mapped strongly negative, and positive inputs will be mapped strongly positive; this symmetric property allows for normalisation of layer's outputs, therefore avoiding using other mechanisms like batch normalisation and allowing our neural network to be less complex. The *sigmoid* function was applied to the last output node to give a value between 0 and 1 to represent the likelihood that the individual has Alzheimer's disease.

Monte Carlo dropout was employed by sampling (i.e. making a forward pass) 50 times from the model, after which a mean and standard deviation were calculated. The mean corresponds to the final model prediction (i.e. likelihood of Alzheimer's disease), and the standard deviation represents the uncertainty of the model. A higher number of samples would bring increased statistical power

to the Bayesian approximation process, but it would also increase the inference time, thus this number (i.e., 50) was chosen as a good compromise in accordance with previous literature[48].

We highlight that a more systematic hyperparameter search could potentially bring better metrics on the ADNI validation set, but we consider such extensive exploration to be beyond the scope of this work, and with diminished returns given the small size of the ADNI dataset.

The model was implemented using Pytorch[51] and trained for 100 epochs using the Adam optimiser[52] with the default learning rate of 0.001. Training convergence was achieved under 50 epochs, therefore 100 epochs for training was considered reasonable. A small weight decay was set to 0.0001 to help with regularisation, and binary cross entropy loss was chosen given the prediction of binary output. The training procedure took 9 s on a server with a TITAN X Pascal GPU and an Intel(R) Core(TM) i7-6900K CPU with 16 cores. The model with the smallest loss on the validation set during the training procedure was selected as the final model for evaluation. Inference time (i.e. 50 forward passes with output calculation) took an average of 12.7 ms (std: 1.78 ms) on GPU (average calculated over 1000 runs for the same batched input). The training log was saved using *Weights & Biases*[53]. In total, the model contained 36,609 trainable parameters.

**Reporting summary**. Further information on research design is available in the Nature Portfolio Reporting Summary linked to this article.

**Statistical analysis**
To assess group differences in the association between AD scores and clinical measures, we used a Bayesian statistical approach given the different sizes of the cohorts used in this study and the limitations of frequentist analysis in identifying statistically significant but clinically irrelevant group differences. We used Stan[54,55] implemented in R (version 4.1.0) using linear regression and logistic regression implemented in the *brms* library[56,57], and the *rstan* library for piecewise linear regression. To assess evidence for group differences we use the Region of Practical Equivalence (ROPE), which is an a priori effect size considered to be significant between groups. The 95% distribution of the Bayesian posterior is termed the critical interval (CI); if the mean lies outside the ROPE there is some evidence to accept a hypothesis between groups, and where the CI lies outside the ROPE there is strong evidence for accepting a hypothesis[58]. The null hypothesis can be accepted where the CI lies completely within the ROPE. The ROPE is either set by knowledge of the

**Table 4 Performance metrics across datasets with a model trained on the ADNI training set, using a cut-off of and Alzheimer's disease (AD) score of 0.5 and employing inference using MC Dropout with 50 samples.**

| Dataset | Accuracy | AUC | Sensitivity | Specificity | PPV/precision | NPV |
|---|---|---|---|---|---|---|
| ADNI test set | 0.92 | 0.97 | 0.90 | 0.93 | 0.90 | 0.93 |
| NACC (only AD/Control) | 0.74 | 0.79 | 0.68 | 0.78 | 0.65 | 0.80 |
| NACC (AD/All) | 0.72 | 0.76 | 0.68 | 0.73 | 0.56 | 0.83 |

*AUC* area under the ROC curve, *PPV* positive predictive value, *NPV* negative predictive value.

variable or set to be 0.1 of the standard deviation of the control group[58]. Model comparison used the *loo* package[59]. To assess the validity of our chosen breakpoint against variable or no breakpoint, we used the expected log pointwise predicted density (ELPD) as the measure of model fit, assessing the difference in ELPD value between models and its standard error to consider whether there was evidence of a difference between models[60].

## Results

**Model evaluation and performance.** Using the ADNI dataset, we trained our deep learning model to detect Alzheimer's disease from structural neuroimaging. To evaluate model performance in the test set of the ADNI cohort and the NACC cohort, we report ROC curve analyses in Table 4. For the NACC dataset, we evaluated AD identification against two comparator groups: (1) controls alone, and (2) combined controls and non-AD diagnoses. As expected, given the similarity to the training set and the selective nature of the cohort, the highest accuracy was found in the ADNI test set. In NACC, a completely independent dataset, accuracy was lower but still reasonable and in line with a previous similar study using SVM for out-of-distribution classification of AD[61]. Overall accuracy was above 0.7, although with a loss in positive predictive value (0.56). Of particular importance, the negative predictive value remained relatively high (0.83). These metrics show that our algorithm is balanced toward missing some people with Alzheimer's disease, but is less likely to label healthy people as having an Alzheimer's disease neuroimaging phenotype. The bias towards a relatively high negative predictive value is relatively better than the reverse situation given the application to UK Biobank data where the rate of Alzheimer's disease will be substantially lower than either ADNI or NACC, so there is a greater risk of misclassifying healthy people as having Alzheimer's disease.

We investigated the relationship between uncertainty measures generated by the model and the predicted value (AD score) in Supplementary Fig. S1a. There was a wider range of uncertainty values when the average AD score was closer to 0.5 than closer to the extremes; in other words, when the probability of classification was greater (towards 0 or 1) the AD score was more certain. Supplementary Fig. S1b demonstrates that when the model prediction was incorrect, its corresponding uncertainty value was higher on average compared to correct predictions. We further compared the Bayesian ML model (including calculation of uncertainty with multiple passes) to a non-stochastic (single pass) one; in our analysis explained in detail in Supplementary Figs. S2–S4, the Bayesian ML model consistently achieved better performance. In Supplementary Figs. S5 and S6, we illustrate possible explainability capacities of our model when using it together with SHapley Additive exPlanations (SHAP)[62], a unified framework for interpreting predictions.

For comparison purposes, we trained the model with the same hyperparameters and preprocessing steps on a balanced training ADNI set (i.e., by having the same number of people in both the AD and control groups). In general, evaluation metrics do not improve on the ADNI and NACC validation sets with this setting, which is expected given the already small training set size (see

Supplementary Table S1). To understand how much our model improves over known risk factors, we fitted a linear regression model to the training (ADNI) set using ordinary least squares (OLS), in which the dependent variable was AD diagnosis, and the independent variables were left hippocampus volume, right hippocampus volume, age, estimated total intracranial volume, and sex. In Supplementary Table S2 it is possible to see that our model is significantly better.

### ADNI

*Clinical scores.* To assess the clinical validity of the AD score, we assessed the difference in clinical scores between those categorised as positive or negative by AD score using a cut-off of 0.5 and applying Bayesian regression models with age as a covariate; the posterior distributions are shown in Fig. 2. Given the distribution of the observed data, skewed Gaussian families were used for Mini-Mental State Examination (MMSE)[63] and Clinical Dementia Rating (CDR) Sum of Boxes[64], otherwise Gaussian distributions were assumed with Cauchy distribution priors in all cases. All models converged well ($\hat{R} \approx 1.00$). We report four key cognitive measures from the ADNI dataset, finding very strong evidence for a difference between AD score positive and negative groups in MMSE (Effect size −5.2, 95% Credible Interval −5.5 to −4.8), Montreal Cognitive Assessment (MoCA)[65] (−8.2, CI −9.1 to −7.4), CDR (3.7, CI 3.5–3.9), and Trails B[66] (−13.0, CI −13.6 to −12.4).

### NACC

*Clinical scores.* We applied the trained model to the NACC datasets and assessed the relation of the model-derived AD-score against clinical scores. Group differences were assessed with Bayesian analysis using the ROPE to assess the strength of evidence, shown in Fig. 3. There was strong evidence that people with a positive AD score had lower MMSE scores (−3.82, CI −4.62 to −3.02), MoCA scores (−7.00, CI −8.33 to −5.69), semantic fluency (−4.64, CI −5.49 to −3.80) and executive function (time taken to complete trails B 44.43, CI 33.36–55.63). For Wechsler Adult Intelligence Score (WAIS)[67] scores (−6.61, CI −9.12 to −4.10) and Boston naming test[68] (−2.97, CI −3.95 to −1.98) there was moderate evidence of a difference in that the mean effect size of the AD score positive group fell outside the ROPE but the critical interval overlapped with the AD score negative, suggesting imprecision in the estimate of the AD score negative group; this may be explained by the relatively low Positive Predictive Value so that some people with Alzheimer's disease are included in the negative AD score group. Finally, there was good evidence that the AD score does not predict forward (−0.19, CI −0.57 to 0.19) or backward (−0.46, CI to −0.86 to −0.06) digit span, given the distribution of the AD score positive scores, is completely contained within the critical interval of the AD score negative group.

To assess whether the severity of the disease was associated with the strength of expression of the AD neuroimaging phenotype, we regressed the AD score against *z*-scored clinical

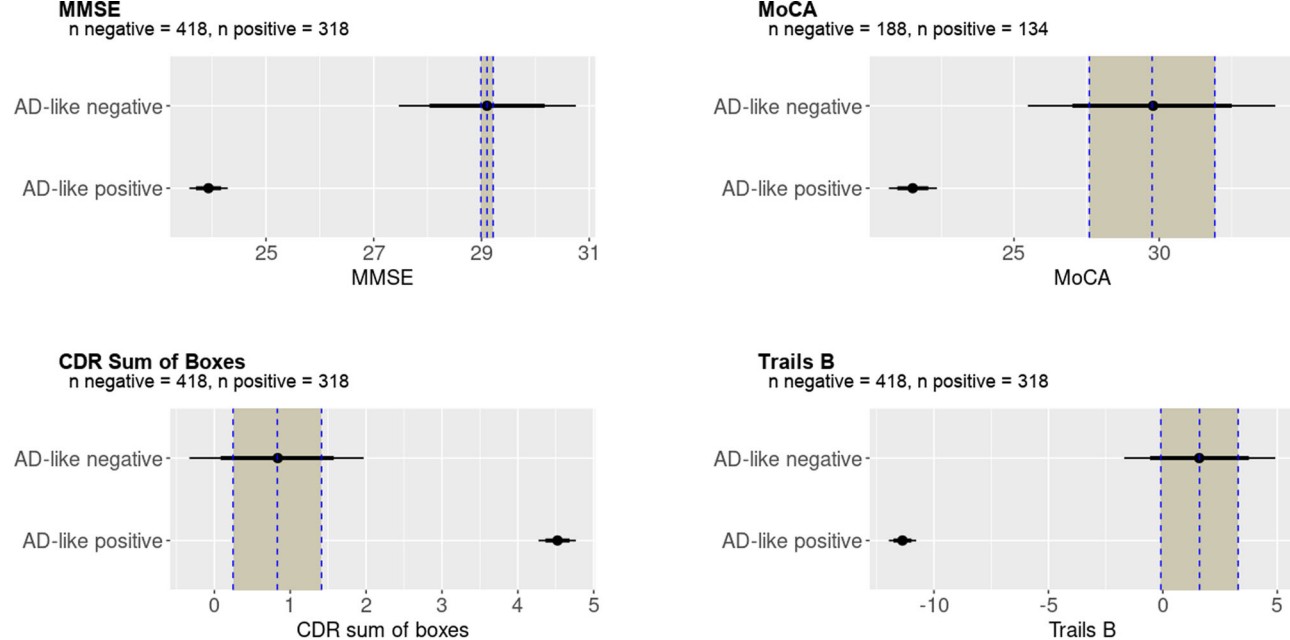

**Fig. 2 Bayesian analysis of cognitive tests in the ADNI dataset.** These distributions represent the Bayesian posterior estimates of the mean of cognitive tests in Alzheimer's disease (AD) score positive and negative groups, with the Region Of Practical Equivalence (ROPE) as a shaded column. As expected in this well-characterised dataset, for all measures there was very strong evidence of a difference between groups classified as positive or negative by AD score derived from structural neuroimaging, indicated by mean AD score in the AD score positive group and the 95% credible intervals (indicated by the thin horizontal bars) falling outside the ROPE. The mean of the AD score negative group is represented by the dotted vertical line with the ROPE denoted by the shaded area on each side. The 75% credible interval is denoted by the thick bars and the 95% credible interval by the thin bars.

measures. We used piecewise linear regression analysis given that we did not expect an association in the AD score negative group (below 0.5) compared with the AD score positive group. Firstly, we assessed whether the piecewise regression model was superior to a linear model, and whether our chosen breakpoint of 0.5 was reasonable by comparing piecewise linear regression models with a fixed breakpoint of 0.5, with variable breakpoint (permitted to vary between 0.25 and 0.75), and with no breakpoint (i.e. completely linear), see Tables S5–S7 in the supplementary data for the full results. The analysis presented in Table 5 shows that models including a breakpoint were superior to the model without a breakpoint for all measures where we found evidence for a difference between the AD score positive and AD score negative groups, specifically MMSE, MoCA, and semantic fluency. There was no substantial difference in whether the breakpoint was fixed at 0.5 or permitted to vary for almost all measures; for the Boston naming task, the variable breakpoint analysis was a better fit than the fixed breakpoint analysis, speculatively because executive cognitive function appears later than other cognitive impairments.

We, therefore, proceeded with our estimated breakpoint of 0.5 to differentiate AD score positive from AD score negative scores, as shown in Fig. 3. There was evidence of a relationship between stronger expression in the AD score positive group than the AD score negative group of the AD score with more impaired cognitive function measured by MMSE, MoCA, forward digit span, trails B and semantic fluency and the Boston naming task, all with a credible interval lying outside the range −0.1 to 0.1 standard deviations of the control group mean.

**UK Biobank.** Using a cut-off for the AD score of 0.5, we divided the UK Biobank cohort into AD score positive or AD score negative groups. There were 1304 (3.4%) with a positive AD score and 36,663 (96.6%) with a negative AD score. All 6 people with a diagnosis of Alzheimer's disease at scan had an AD score >0.5

(range 0.60–0.95) and were not included in any subsequent analysis. The group with a positive AD score was only slightly older than the AD score negative group (1.79 years, CI 1.39–2.21).

*AD scores predict cognitive differences in healthy individuals with an AD imaging phenotype.* To assess for differences in cognitive scores between the groups, we used Bayesian linear or logistic regression models. All models achieved good convergence ($\widehat{R} \approx 1$) and the results are shown in Fig. 4.

There was strong evidence of worse fluid intelligence in the AD score positive group (−0.35, CI −0.46 to −0.21) with the 95% CI lying completely outside the ROPE. There was moderate evidence to support poorer performance in matrix pattern completion (−0.35, CI −0.50 to −0.20), numeric memory (−0.17, CI −0.27 to −0.07), and reaction time for correct trials (13.11 ms, CI 7.12–19.33 ms), where the mean estimate was outside the ROPE, but the CI overlapped with the ROPE. On a working memory task (pairs matching) there was only weak evidence to suggest a poorer performance in the AD score-positive groups performance using logistic regression with an adjacent categories model; with AD score-positive participants slightly more likely to have one rather than two correct answers out of four (boundary effect size −0.14, CI −1.02 to 0.65), and slightly more likely to have two rather than three correct answers (boundary effect size −1.74, I −5.19 to 0.79), and slightly more likely to have three rather than four correct answers (boundary effect size 0.95, CI −0.36 to 3.46). There was also weak evidence to suggest poorer performance on a prospective memory task (increased probability probably of an incorrect answer in the AD score positive group 0.09, CI 0.00–0.18).

On tests of executive function, there was clear evidence of no difference in the number of errors on the Trails B test (0.12, CI −0.24 to 0.48) where the credible interval was completely within the ROPE, and weak evidence against an effect in tower rearranging (−0.31, CI −0.53 to −0.08) where the mean lies within the ROPE but the credible interval extends beyond the ROPE.

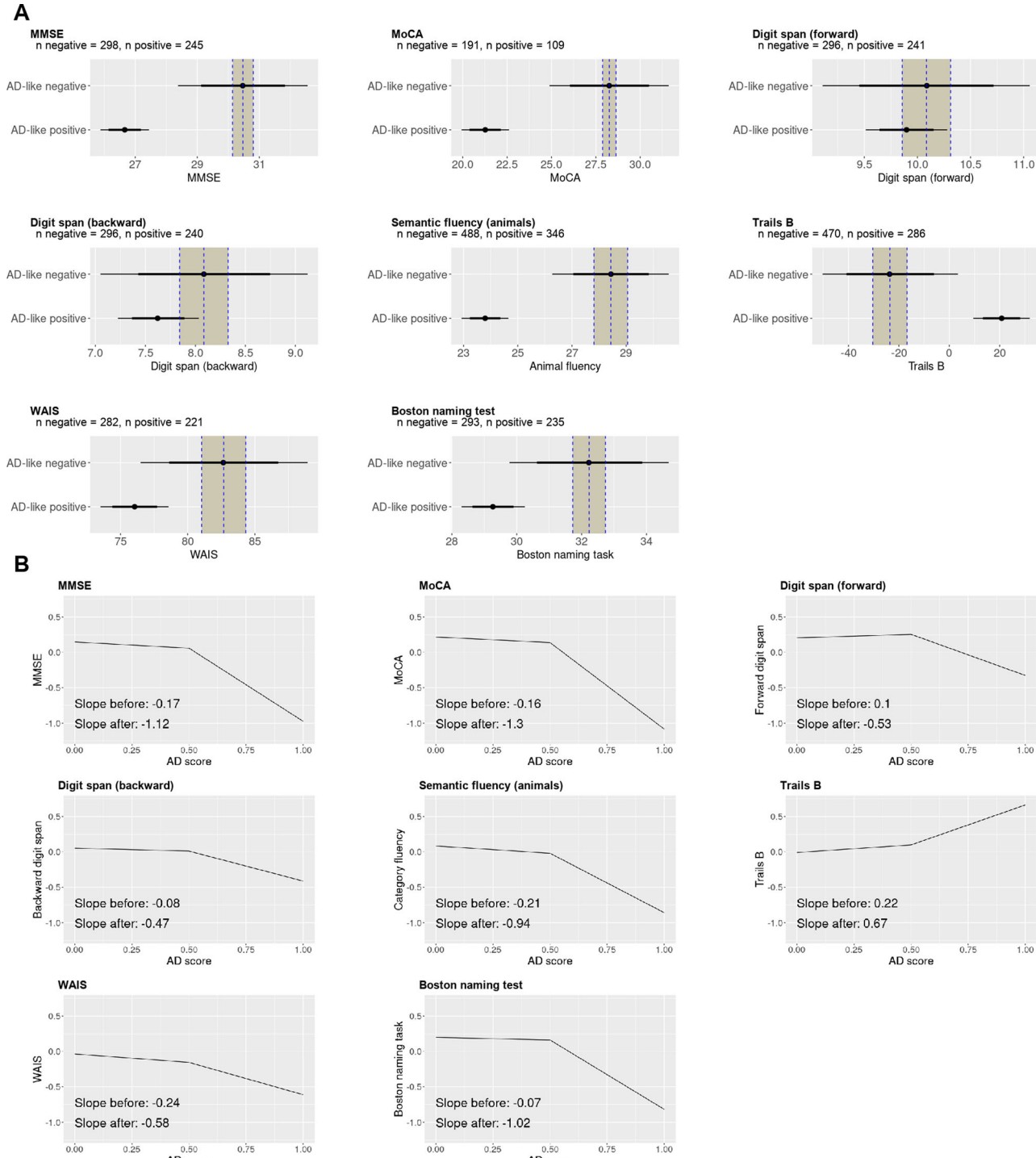

**Fig. 3 Analyses of the NACC clinical scores. A** Bayesian analysis of the NACC clinical scores. There is strong evidence for impairment in the Alzheimer's disease (AD) score positive group for Mini-Mental State Examination (MMSE), Montreal Cognitive Assessment (MoCA), and trails B since the posterior estimate of the effect size lies outside the 95% credible interval, and outside the Region Of Practical Equivalence (ROPE). There is good evidence of no difference for forward and backward digit span since in both cases the distribution of the AD score positive group completely overlaps with the distribution of the AD score negative group. The mean of the AD score negative group is represented by the dotted vertical line with the ROPE denoted by the shaded area on each side. The 75% credible interval is denoted by the thick bars and the 95% credible interval by the thin bars. **B** Breakpoint analysis of the NACC clinical scores. Disease severity correlated with the AD score positive group (AD score > 0.5) with evidence for a difference in correlation from the AD negative (AD score < 0.5) group in MMSE, MoCA, forward digits span, Trails B, and the Boston naming task.

*AD score predicts worse reported overall health.* In non-cognitive measures, there was strong evidence that people in the AD group were more likely to report their overall health as 'poor' or 'fair' rather than 'good' or 'excellent' (see Fig. 5) (probit 0.14, CI 0.09–0.19). There was weak evidence that hand grip was weaker in the AD score positive group with a mean outside the ROPE but the CI overlapping with the ROPE (mean −1.10, CI −1.70 to −0.51). There was also weak evidence that the AD score positive

**Table 5 Here, we test the cut-off Alzheimer's disease (AD) score value of 0.5 in the NACC dataset by applying linear analysis of the relationship between AD score and cognitive scores using Bayesian piecewise linear regression analysis**

| | **Variable breakpoint** | | | | |
|---|---|---|---|---|---|
| | **BP** | **Slope < BP (CI)** | **Slope > BP (CI)** | **Slope diff (CI)** | **ELPD diff (se)** |
| MMSE | 0.67 | −0.19 (−1.8, −0.94) | **−1.4 (−1.8, −0.94)** | **−1.2 (−1.7, −0.65)** | 0.0 (0.0) |
| MoCA | 0.6 | −0.18 (−0.35, 0.03) | **−1.40 (−1.8, −1.0)** | **−1.2 (−1.8, −0.77)** | 0.0 (0.0) |
| Backward digit span | 0.56 | −0.08 (−0.25, 0.08) | **−0.51 (−0.96, −0.13)** | −0.43 (−1.0, 0.10) | 0.0 (0.0) |
| Forward digit span | 0.59 | 0.09 (−0.08, 0.26) | **−0.62 (−1.1, −0.2)** | **−0.71 (−1.3, −0.16)** | 0.0 (0.0) |
| Semantic fluency | 0.59 | **−0.22 (−0.34, −0.10)** | **−1.0 (−1.3, −0.74)** | **−0.8 (−0.18, −0.44)** | 0.0 (0.0) |
| Trails B | 0.44 | 0.2 (0.079, 0.35) | **0.66 (0.45, 0.89)** | **0.46 (0.13, 0.79)** | 0.0 (0.0) |
| WAIS | 0.53 | −0.24 (−0.42, −0.07) | **−0.59 (−0.97, −0.25)** | −0.35 (−0.82, 0.13) | −0.1 (0.1) |
| Boston naming task | 0.66 | −0.08 (−0.23, 0.056) | **−1.3 (−1.7, −0.81)** | **−1.2 (−1.70, −0.64)** | 0.0 (0.0) |
| | **Fixed breakpoint (0.5)** | | | | |
| | **BP** | **Slope < BP (CI)** | **Slope > BP (CI)** | **Slope diff (CI)** | **ELPD diff (se)** |
| MMSE | [0.5] | −0.17 (−0.32, −0.02) | **−1.12 (−1.4, −0.82)** | **−0.95 (−1.4, −0.53)** | −1.3 (1.0) |
| MoCA | [0.5] | −0.16 (−0.34, 0.03) | **−1.3 (−1.6, −1.0)** | **−1.1 (−1.6, −0.72)** | −0.5 (0.6) |
| Backward digit span | [0.5] | −0.08 (−0.26, 0.09) | **−0.47 (−0.81, −0.13)** | −0.38 (−0.87, 0.10) | −0.3 (0.2) |
| Forward digit span | [0.5] | 0.10 (−0.07, 0.28) | **−0.53 (−0.87, −0.2)** | **−0.63 (−1.1, −0.15)** | −0.3 (0.3) |
| Semantic fluency | [0.5] | −0.21 (−0.34, −0.08) | **−0.94 (−1.20, -0.72)** | **−0.74 (−1.1, −0.41)** | −0.5 (0.4) |
| Trails B | [0.5] | 0.22 (0.08, 0.36) | **0.67 (0.46, 0.89)** | **0.46 (0.13, 0.77)** | 0.0 (0.2) |
| WAIS | [0.5] | −0.24 (−0.40, −0.07) | **−0.58 (−0.87, −0.24)** | −0.34 (−0.79, 0.13) | 0.0 (0.0) |
| Boston naming task | [0.5] | −0.07 (−0.23, 0.09) | **−1.0 (−1.3, −0.7)** | **−0.94 (−1.4, −0.49)** | −1.7 (0.8) |

| | **Linear model (no breakpoint)** | |
|---|---|---|
| | **Slope (CI)** | **ELPD diff (se)** |
| MMSE | **−0.47 (−0.55, −0.39)** | **−10.0 (5.3)** |
| MoCA | **−0.60 (−0.69, −0.51)** | **−12.4 (6.8)** |
| Backward digit span | **−0.2 (−0.29, −0.12)** | −0.5 (1.6) |
| Forward digit span | −0.1 (−0.18, −0.02) | −2.6 (2.6) |
| Semantic fluency | **−0.46 (−0.52, −0.40)** | **−9.6 (4.5)** |
| Trails B | **0.39 (0.36, 0.45)** | −2.7 (2.8) |
| WAIS | **−0.35 (−0.43, −0.26)** | −0.3 (1.4) |
| Boston naming task | **−0.37 (−0.45, −0.29)** | **−9.6 (4.7)** |

For the slope estimates, we include the 95% Credible Interval (CI). Given that the data are z-scored, we use 0.1 as the Region of Practical Equivalence (ROPE) which represent 0.1 of the standard deviation of the data. If the CI lies outside −0.1 to 0.1, we consider there is good evidence of a relationship between the AD score and clinic score, indicated by values in bold. For models with variable breakpoints, the breakpoint values obtained were similar to 0.5, and comparing models with the Expected Log Pointwise Predicted Density (ELPD) the difference between variable and fixed breakpoint models was negligible, except for the Boston naming task. There was good evidence for a breakpoint in MMSE, MoCA, forward digit span, semantic fluency, and the Boston naming task, and further evidence supporting no difference between the two breakpoint models; however, both being superior to the non-breakpoint model. These findings support our use of a breakpoint of 0.5.
*BP* breakpoint, *MMSE* Mini Mental State Examination, *MoCA* Montreal Cognitive Assessment, *WAIS* Wechsler Adult Intelligence Scale.

group was more likely to report one fall than no falls and more likely to report two or more falls than no falls (probit regression 0.07, CI 0.06–0.08).

*AD scores are associated with modifiable risk factors.* Having identified a cohort potentially at risk of Alzheimer's disease, the next step was to consider whether other health measures or modifiable risk factors are more common in this subgroup. We report the results of a number of risk factors in Fig. 6 and other health markers in Fig. 5.

There was some evidence of a difference in both diastolic blood pressure (1.12, CI 0.53–1.72) and systolic blood pressure (2.29, CI 1.26–3.30). Additionally, there was weak evidence that smoking (current or ex-smoker) was associated with AD score positive score (0.06, CI −0.06 to 0.18), demonstrating a mean outside the ROPE, but a wide CI. Among those who smoked, there was moderate evidence that a greater smoking history (i.e. more pack years) was associated with an AD score positive score (2.98, CI 1.23–4.73).

There was moderately strong evidence for no difference in waist circumference (0.62, CI −0.11 to 1.35), consultation with GP for depression (logistic regression 0.03, CI −0.10 to 0.16), consultation with a psychiatrist for depression (logistic regression 0.02, CI −0.19 to 0.23), hearing difficulties (logistic regression

−0.01, CI −0.14 to 0.12). There was strong evidence of no difference in hip circumference (−0.12, CI −0.63 to 0.38), sleep duration (−0.01 h, CI 0.07–0.06), and neuroticism (0.11, −0.09 to 0.32) score.

## Discussion

We have identified a cohort of healthy individuals in the UK Biobank with an Alzheimer's disease-like neuroimaging-based intermediate phenotype, by leveraging developments in Bayesian deep learning. Despite having no diagnosis or reported symptoms of dementia at the time of assessment, this AD-like cohort demonstrates a cognitive profile in keeping with early Alzheimer's disease and reports worse general health. In addition, they have evidence of slightly higher blood pressure and longer smoking history as potentially modifiable risk factors.

Our approach offers the opportunity to identify and study presymptomatic idiopathic Alzheimer's disease. The search for the earliest possible changes in Alzheimer's disease has mainly focused on genetic forms of dementia[4,14], with neuroimaging changes in presymptomatic genetic Alzheimer's disease described since the 1990s using PET[69] or structural MRI[70]. The earliest studies to identify prediagnostic structural brain changes of Alzheimer's disease found group-level differences in medial

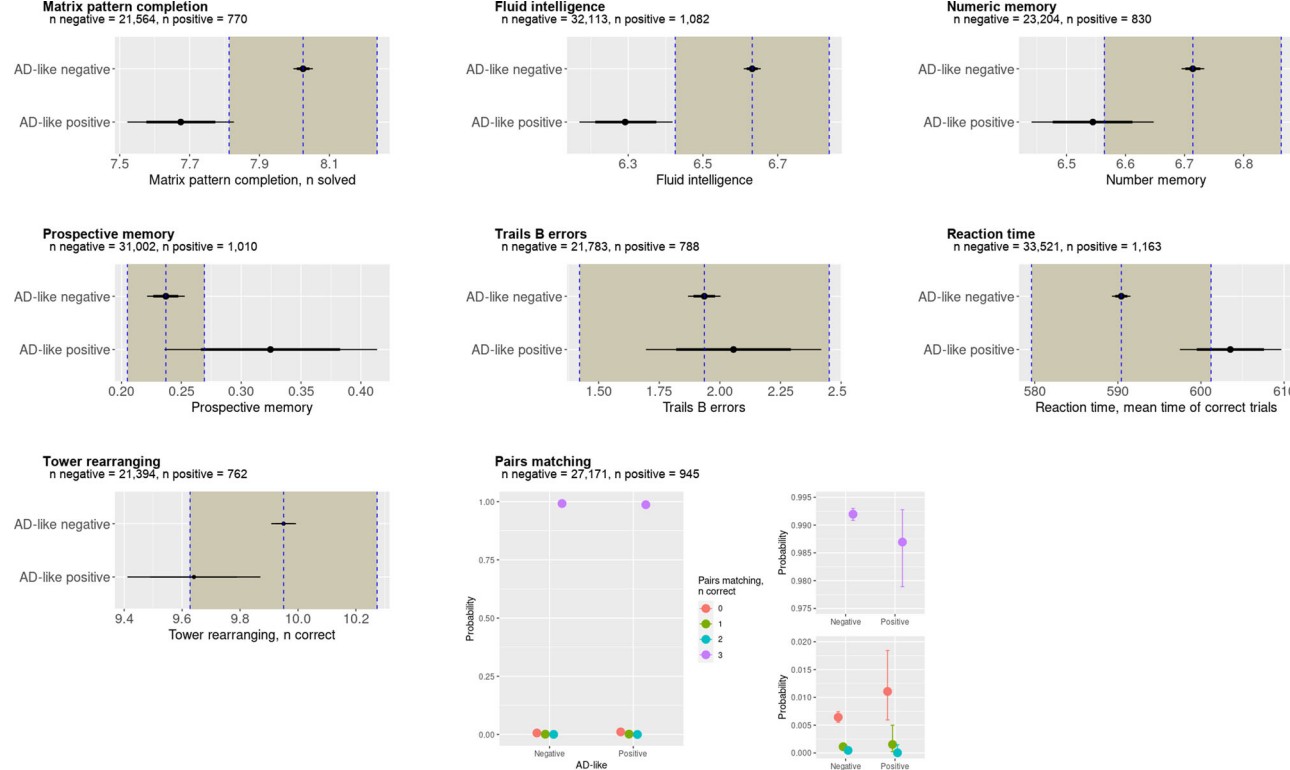

**Fig. 4 Bayesian analysis of cognitive tests in the UK Biobank.** It is possible to see a reduced cognitive function in participants with an Alzheimer's disease (AD) score > 0.5. In particular, there is strong evidence for impaired visual memory, good evidence for impaired fluid intelligence, numeric memory, and some evidence for impaired executive function. The shaded area represents the Region Of Practical Equivalence (ROPE)—if the distribution of the AD-positive group lies outside the ROPE there is strong evidence for a difference between the groups, and if the mean only lies outside the ROPE then there is some to good evidence for a group difference. There was strong evidence for no difference between groups in errors on the trials B task or reaction time, where the distributions lie completely inside the ROPE. For the pairs matching task, we used Bayesian ordinal regression, plotting here the 95% credible intervals demonstrating only weak evidence of fewer correct answers in the AD score positive group. The mean of the AD score negative group is represented by the dotted vertical line with the ROPE denoted by the shaded area on each side. The 75% credible interval is denoted by the thick bars and the 95% credible interval by the thin bars.

temporal structures[6,7], but with low sensitivity (78%) and specificity (75%)[8]. A recent study using ADNI data in 32 people who progressed to MCI and 8 to AD confirmed group-level structural changes in medial temporal structures that were detectable 10 years prior to onset. We are aware of one promising study in idiopathic Alzheimer's disease using a machine learning approach with multimodal imaging data to try to predict individualised presymptomatic disease in the ADNI cohort, currently in preprint[71], an approach that will need independent validation. A study of cognitively normal adults over 70 years of age attempted to detect presymptomatic Alzheimer's disease using FDG-PET, suggesting two-thirds of people in this age group had an abnormal FDG-PET scan which was associated with psychiatric symptoms[72]. This proportion of patients seems high for the age group under consideration, and abnormalities on PET have been associated with depression[73], so the relevance of these findings is unclear. In a small study using Pittsburgh Compound B (PiB) PET to detect presymptomatic Alzheimer's disease in a healthy and MCI cohort, there was a correlation between β-amyloid load and poorer episodic memory, though only one person converted to Mild Cognitive Impairment[74]. Another much larger study found a high rate of positive β-amyloid PET scans in otherwise cognitively normal older adults and no association with cognition, so whilst PET may be helpful for risk stratification, the role and timing of β-amyloid PET abnormalities remain uncertain in the detection of presymptomatic Alzheimer's disease in a clinical setting[75].

In this context, our approach has improved on previous efforts by identifying individuals with possible early sporadic AD, a supposition that is supported by finding a cognitive profile in keeping with AD. Our findings are strengthened by identifying strong correlations between the AD scores and relevant cognitive tests in the independent NACC study. We found that the AD score was associated with worse performance on global cognitive tests such as the MMSE and MoCA, and on more AD-specific cognitive domains of memory and semantic fluency. In the UK Biobank cohort, the AD score was associated with key cognitive domains of AD including memory and fluid intelligence.

Regarding the prospect for disease prevention, our results suggest that smoking history, particularly a greater pack-year history, and both systolic and diastolic hypertension are risk factors. Both smoking and hypertension are reported as risk factors in the 2020 Lancet Commission on Dementia[76]. Smoking is a particularly well-established risk factor for dementia[77]. In keeping with our findings, Rusanen et al.[78] studied over 21,000 people, finding that heavy smoking in middle age was associated with developing Alzheimer's disease and, more specifically, that greater cigarette use was associated with a higher risk of developing dementia. Our results suggest that the effect of smoking is mediated through structural volume loss in key brain regions.

The difference between blood pressure in the AD score positive and AD negative groups was small, ~2.5 mmHg for systolic BP

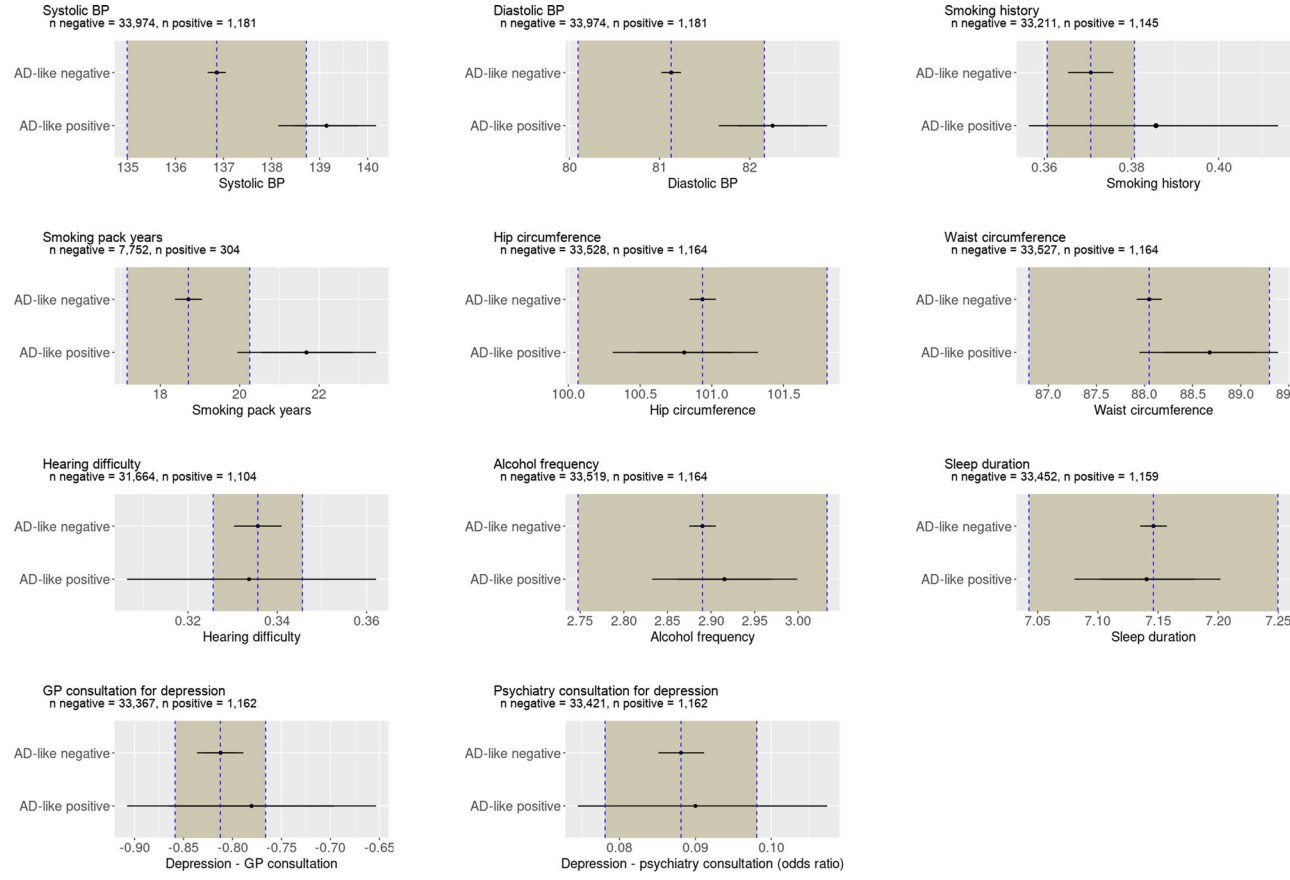

**Fig. 5 Other measures of health from the UK Biobank.** People with positive AD scores were more likely to report their general health to be `fair' or `poor' and less likely to report their general health as `good' or `excellent'. In addition, they had lower grip strength which has previously been associated with Alzheimer's disease. There was weak evidence to suggest that people with a positive AD score were more likely to have had one or more falls in the previous year. The mean of the AD score negative group is represented by the dotted vertical line with the ROPE denoted by the shaded area on each side. The 75% credible interval is denoted by the thick bars and the 95% credible interval by the thin bars.

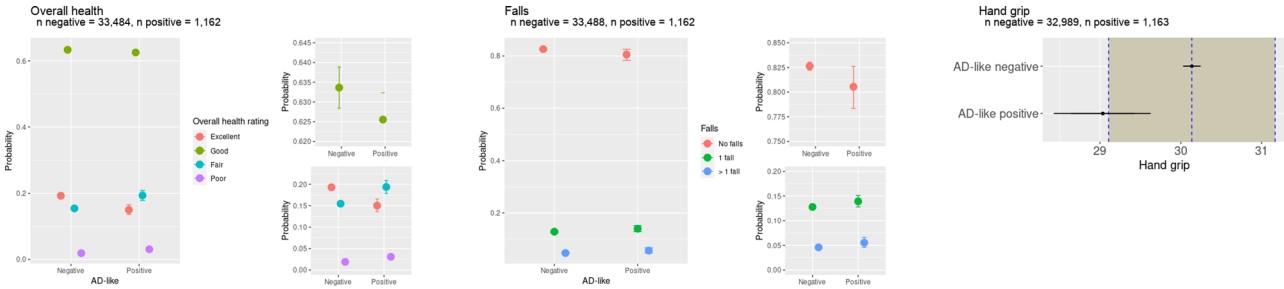

**Fig. 6 Results from Bayesian analysis of potentially modifiable risk factors in the UK Biobank population.** There is partial evidence to support a higher diastolic and systolic blood pressure among participants with an Alzheimer's disease (AD) score > 0.5, indicated by a mean effect size lying outside the Region Of Practical Equivalence (ROPE) but with a distribution overlapping with the ROPE. No other risk factors were associated with a positive AD score. The mean of the AD score negative group is represented by the dotted vertical line with the ROPE denoted by the shaded area on each side. The 75% credible interval is denoted by the thick bars and the 95% credible interval by the thin bars.

and 1 mmHg for diastolic BP. There has been much debate on the relationship between blood pressure and cognitive impairment, with studies finding both high and low diastolic blood pressure to be related to Alzheimer's disease[79,80]. More recent evidence from a meta-analysis has suggested that mid-life hypertension is a greater risk factor, with a systolic blood pressure above 140 mmHg conferring a relative risk of 1.2 for developing dementia, and systolic blood pressure above 80 mmHg conferring a relative risk of 1.54[81]. However, the small increase in blood pressure we identified in the AD score positive group, and the overlap with the AD score negative group in both systolic and

diastolic blood pressures suggests heterogeneity within the AD score positive group.

We did not find differences in other potentially modifiable risk factors. Here, the Bayesian approach is helpful since we can confidently reject the possibility of some risk factors being associated with the AD neuroimaging phenotype in this group. For example, some of the risk factors highlighted in the Lancet Commission 2020 report[76], were not identified as risks in the current study (i.e. alcohol frequency, hip circumference, sleep duration) since the distribution of the AD score positive group lies wholly within the ROPE (see Fig. 6). For depression and

hearing difficulty, there was a wide distribution of estimated risk beyond the ROPE suggesting an imprecise estimate of the risk. We cannot rule out an association with an AD neuroimaging phenotype for these measures.

Two factors may have limited our ability to identify potentially modifiable risk factors. Firstly, the UK Biobank has a sample bias towards people who are healthier with fewer disease risk factors than the general UK population[82]. For example, the proportion of people currently smoking in the UK Biobank population is 10.7% compared to 14.7% in the general population (data from the Office for National Statistics[83]. Moreover, the restricted mid-life age range of the UK Biobank cohort may exclude the age at which some risk factors apply most strongly.

Secondly, our model was biased towards a high negative predictive value, meaning that we may have 'missed' some people with early Alzheimer's disease pathology. Whilst providing more confidence in the identification of an AD-like cohort, the potential classification of people with latent AD in the AD-negative group may have reduced the power to detect a difference in risk factors between the AD score positive and AD-negative groups. We anticipate that combining neuroimaging with other risk biomarkers could improve the selection of a high-risk group, for example, blood biomarkers[84,85] or polygenic risk scores[86,87].

Despite these caveats, this approach has the potential to enrich dementia prevention trials. It is important to note that the impact of addressing risk factors on preventing dementia is not yet well established. The World Wide FINGERS study has reported a trial of a multi-domain intervention with a small but significant effect size[1], although this was not targeted at smoking cessation or lowering blood pressure specifically, and there was no difference in blood pressure between the intervention and control groups at the end of the study. Our findings support the need for such trials but raise some caution about the prevalence and strength of the association between risk factors and AD pathology.

To identify the AD score positive group we used a state-of-the-art Bayesian ML approximation method (i.e. Monte Carlo dropout[47]) to identify the cohort of interest in the UK Biobank. The Bayesian approach allows a model to predict not only a single AD-likelihood value as in typical deterministic neural networks but also a measure of uncertainty (see Fig. 1). A key advantage of this approach is the additional information about the generalisability of the model to challenging out-of-distribution datasets, such as we have done in this paper; for example, we were able to identify that greater uncertainty was associated with incorrect predictions (see supplementary Fig. S1b).

Our approach is particularly well-validated compared to other similar models. The model was trained only on the ADNI dataset before validation on the completely independent and significantly more noisy NACC data, prior to application to the UK Biobank. All the confound corrections on the input data were conducted in the training dataset (i.e. ADNI) alone, and correction statistics are then applied to the external datasets; in this way, we avoid biases that would have been introduced had we corrected the model on all the available data.

There are limitations to our approach. Most importantly, we do not know at present whether the people identified as having a positive AD score will go on to develop the syndrome of Alzheimer's disease. At the time of analysis, only 17 people in the neuroimaging cohort have developed dementia (6 of these self-reported at the baseline visit). The neuroimaging sub-study began later than the main biobank study, so it may be some years before a sizeable population of people with dementia and neuroimaging is available. An ideal dataset for training our model would be comprised of people prior to a diagnosis of Alzheimer's disease,

which does not yet exist in sufficient size to train a deep learning model. Despite these caveats, we propose that the group we identified from their AD-like imaging phenotype is at higher risk of future clinical Alzheimer's disease.

Whilst our model performed very well in the ADNI population in which it was trained, it performed, as expected, less well in the independent NACC population. There are a number of reasons for this. Firstly, there is a recognised selection bias when using the ADNI cohort which may lead to an overly optimistic classification[39]. Secondly, the NACC dataset relies on clinical diagnosis rather than a defined set of diagnostic criteria without pathological information or biomarkers such as CSF; therefore, a lower diagnostic accuracy might be expected. Thirdly, the neuroimaging quality varies significantly in the NACC dataset; for example, both 1.5 and 3 T MRI scans were included. For these reasons, it is not surprising that the classification was poorer in the NACC dataset, though still with good metrics for the task at hand.

Using Bayesian statistics for group comparison and regression models provided several clear advantages for this study. Firstly, given the unequal sizes of the positive and negative groups we were able to focus on the precision of parameter estimates given the available data which differed between the two groups; this meant that we could distinguish a small effect size from an imprecise parameter estimate. Secondly, we were able to use effect size to detect evidence of difference between groups; if we had used a traditional frequentist approach we would have had difficult choices about correction for multiple comparisons and concern about detecting small but clinically irrelevant differences. Finally, using Bayesian analysis enabled us to explicitly accept the null hypothesis (i.e. no difference between groups) in a number of statistical comparisons.

In conclusion, we demonstrate an approach to identify a cohort of potentially presymptomatic sporadic Alzheimer's disease using AI with structural neuroimaging to identify a neuroimaging phenotype.

## Data availability
All the data used in this study is available by application to the data managers of ADNI (www.adni-info.org), NACC (https://naccdata.org/), and the UK Biobank (https://www.ukbiobank.ac.uk/). ADNI data used in the preparation of this article were obtained from the Alzheimer's Disease Neuroimaging Initiative (ADNI) database (adni.loni.usc.edu). The ADNI was launched in 2003 as a public–private partnership, led by Principal Investigator Michael W. Weiner, MD. The primary goal of ADNI has been to test whether serial magnetic resonance imaging (MRI), positron emission tomography (PET), other biological markers, and clinical and neuropsychological assessment can be combined to measure the progression of mild cognitive impairment (MCI) and early Alzheimer's disease (AD). For up-to-date information, see www.adni-info.org. The patient-level original and preprocessed data cannot be directly shared due to restrictions set by each consortium (i.e., ADNI, NACC, and the UK Biobank). The outputs of the deep learning model used to generate the figures can be accessed in CSV format in the same GitHub repository, under the folder "results/".

## Code availability
Code (for reproducibility) and results are publicly available on GitHub, with additional instructions for implementation: https://github.com/tjiagoM/adni_phenotypes[88].

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

## Acknowledgements

T.A. was funded by the W.D. Armstrong Trust Fund, University of Cambridge, UK. T.R. and J.B.R. are supported by the Cambridge Centre for Parkinson-plus and NIHR Cambridge Biomedical Research Centre (BRC-1215-20014). The views expressed are those of the authors and not necessarily those of the NIHR or the Department of Health and Social Care. J.B.R. is supported by the Medical Research Council (SUAG/051 R101400) and Wellcome Trust (103838). P.L. is supported by funding from the EU GOD-DS21 scheme (Grant agreement No. 848077). Data collection and sharing for this project was funded by the Alzheimer's Disease Neuroimaging Initiative (ADNI) (National Institutes of Health Grant U01 AG024904) and DOD ADNI (Department of Defense award number W81XWH-12-2-0012). ADNI is funded by the National Institute on Aging, the National Institute of Biomedical Imaging and Bioengineering, and through generous contributions from the following: AbbVie, Alzheimer's Association; Alzheimer's Drug Discovery Foundation; Araclon Biotech; BioClinica, Inc.; Biogen; Bristol-Myers Squibb Company; CereSpir, Inc.; Cogstate; Eisai Inc.; Elan Pharmaceuticals, Inc.; Eli Lilly and Company; EuroImmun; F. Hoffmann-La Roche Ltd and its affiliated company Genentech, Inc.; Fujirebio; GE Healthcare; IXICO Ltd.; Janssen Alzheimer Immunotherapy Research & Development, LLC.; Johnson & Johnson Pharmaceutical Research & Development LLC.; Lumosity; Lundbeck; Merck & Co., Inc.; Meso Scale Diagnostics, LLC.; NeuroRx Research; Neurotrack Technologies; Novartis Pharmaceuticals Corporation; Pfizer Inc.; Piramal Imaging; Servier; Takeda Pharmaceutical Company; and Transition Therapeutics. The Canadian Institutes of Health Research is providing funds to support ADNI clinical sites in Canada. Private sector contributions are facilitated by the Foundation for the National Institutes of Health (www.fnih.org). The grantee organization is the Northern California Institute for Research and Education, and the study is coordinated by the Alzheimer's Therapeutic Research Institute at the University of Southern California. ADNI data are disseminated by the Laboratory for Neuro Imaging at the University of Southern California. The NACC database is funded by NIA/NIH Grant U24 AG072122. NACC data are contributed by the NIA-funded ADCs: P50 AG005131 (PI James Brewer, MD, PhD), P50 AG005133 (PI Oscar Lopez, MD), P50 AG005134 (PI Bradley Hyman, MD, PhD), P50 AG005136 (PI Thomas Grabowski, MD), P50 AG005138 (PI Mary Sano, PhD), P50 AG005142 (PI Helena Chui, MD), P50 AG005146 (PI Marilyn Albert, PhD), P50 AG005681 (PI John Morris, MD), P30 AG008017 (PI Jeffrey Kaye, MD), P30 AG008051 (PI Thomas Wisniewski, MD), P50 AG008702 (PI Scott Small, MD), P30 AG010124 (PI John Trojanowski, MD, PhD), P30 AG010129 (PI Charles DeCarli, MD), P30 AG010133 (PI Andrew Saykin, PsyD), P30 AG010161 (PI David Bennett, MD), P30 AG012300 (PI Roger Rosenberg, MD), P30 AG013846 (PI Neil Kowall, MD), P30 AG013854 (PI Robert Vassar, PhD), P50 AG016573 (PI Frank LaFerla, PhD), P50 AG016574 (PI Ronald Petersen, MD, PhD), P30 AG019610 (PI Eric Reiman, MD), P50 AG023501 (PI Bruce Miller, MD), P50 AG025688 (PI Allan Levey, MD, PhD), P30 AG028383 (PI Linda Van Eldik, PhD), P50 AG033514 (PI Sanjay Asthana, MD, FRCP), P30 AG035982 (PI Russell Swerdlow, MD), P50 AG047266 (PI Todd Golde, MD, PhD), P50 AG047270 (PI Stephen Strittmatter, MD, PhD), P50 AG047366 (PI Victor Henderson, MD, MS), P30 AG049638 (PI Suzanne Craft, PhD), P30 AG053760 (PI Henry Paulson, MD, PhD), P30 AG066546 (PI Sudha Seshadri, MD), P20 AG068024 (PI Erik Roberson, MD, PhD), P20 AG068053 (PI Marwan Sabbagh, MD), P20 AG068077 (PI Gary Rosenberg, MD), P20 AG068082 (PI Angela Jefferson, PhD), P30 AG072958 (PI Heather Whitson, MD), P30 AG072959 (PI James Leverenz, MD). This research has been conducted using data from UK Biobank, a major biomedical database (http://www.ukbiobank.ac.uk/).

## Author contributions

T.A. conducted the experiments, T.A., J.R., and T.R. conceived the experiments and analysed the results, R.A.I.B. carried out neuroimaging preprocessing and analysis, D.W. and N.S. helped with statistical analysis, T.R. and P.L. supervised the study. All authors reviewed the manuscript. Data used in the preparation of this article were obtained from the Alzheimer's Disease Neuroimaging Initiative (ADNI) database (adni.loni.usc.edu). As such, the investigators within the ADNI contributed to the design and implementation of ADNI and/or provided data but did not participate in the analysis or writing of this report. A complete listing of ADNI investigators can be found at: http://adni.loni.usc.edu/wp-content/uploads/how_to_apply/ADNI_Acknowledgment_List.pdf.

## Competing interests

The authors declare no competing interests.

## Additional information

## the Alzheimer's Disease Neuroimaging Initiative

Lisa C. Silbert[5], Betty Lind[5], Rachel Crissey[5], Jeffrey A. Kaye[5], Raina Carter[5], Sara Dolen[5], Joseph Quinn[5], Lon S. Schneider[6], Sonia Pawluczyk[6], Mauricio Becerra[6], Liberty Teodoro[6], Karen Dagerman[6], Bryan M. Spann[6], James Brewer[7], Helen Vanderswag[7], Adam Fleisher[7], Jaimie Ziolkowski[8], Judith L. Heidebrink[8], Zbizek Nulph[8], Joanne L. Lord[8], Lisa Zbizek-Nulph[8], Ronald Petersen[9], Sara S. Mason[9], Colleen S. Albers[9], David Knopman[9], Kris Johnson[9], Javier Villanueva-Meyer[10], Valory Pavlik[10], Nathaniel Pacini[10], Ashley Lamb[10], Joseph S. Kass[10], Rachelle S. Doody[10], Victoria Shibley[10], Munir Chowdhury[10], Susan Rountree[10], Mimi Dang[10], Yaakov Stern[11], Lawrence S. Honig[11], Akiva Mintz[11], Beau Ances[12], John C. Morris[12], David Winkfield[12], Maria Carroll[12], Georgia Stobbs-Cucchi[12], Angela Oliver[12], Mary L. Creech[12], Mark A. Mintun[12], Stacy Schneider[12], David Geldmacher[13], Marissa Natelson Love[13], Randall Griffith[13], David Clark[13], John Brockington[13], Daniel Marson[13], Hillel Grossman[14], Martin A. Goldstein[14], Jonathan Greenberg[14], Effie Mitsis[14], Raj C. Shah[15], Melissa Lamar[15], Ajay Sood[15], Kimberly S. Blanchard[15], Debra Fleischman[15], Konstantinos Arfanakis[15], Patricia Samuels[15], Ranjan Duara[16], Maria T. Greig-Custo[16], Rosemarie Rodriguez[16], Marilyn Albert[17], Daniel Varon[17], Chiadi Onyike[17], Leonie Farrington[17], Scott Rudow[17], Rottislav Brichko[17], Maria T. Greig[17], Stephanie Kielb[17], Amanda Smith[18], Balebail Ashok Raj[18], Kristin Fargher[18], Martin Sadowski[19], Thomas Wisniewski[19], Melanie Shulman[19], Arline Faustin[19], Julia Rao[19], Karen M. Castro[19], Anaztasia Ulysse[19], Shannon Chen[19], Mohammed O. Sheikh[19], Jamika Singleton-Garvin[19], P. Murali Doraiswamy[20], Jeffrey R. Petrella[20], Olga James[20], Terence Z. Wong[20], Salvador Borges-Neto[20], Jason H. Karlawish[21], David A. Wolk[21], Sanjeev Vaishnavi[21], Christopher M. Clark[21], Steven E. Arnold[21], Charles D. Smith[22], Gregory A. Jicha[22], Riham El Khouli[22], Flavius D. Raslau[22], Oscar L. Lopez[23], Michelle Zmuda[23], Meryl Butters[23], MaryAnn Oakley[23], Donna M. Simpson[23], Anton P. Porsteinsson[24], Kim Martin[24], Nancy Kowalski[24], Kimberly S. Martin[24], Melanie Keltz[24], Bonnie S. Goldstein[24], Kelly M. Makino[24], M. Saleem Ismail[24], Connie Brand[24], Christopher Reist[25], Gaby Thai[25], Aimee Pierce[25], Beatriz Yanez[25], Elizabeth Sosa[25], Megan Witbracht[25], Brendan Kelley[26], Trung Nguyen[26], Kyle Womack[26], Dana Mathews[26], Mary Quiceno[26], Allan I. Levey[27], James J. Lah[27], Ihab Hajjar[27], Janet S. Cellar[27], Jeffrey M. Burns[28], Russell H. Swerdlow[28], William M. Brooks[28], Daniel H. S. Silverman[29], Sarah Kremen[29], Liana Apostolova[29], Kathleen Tingus[29], Po H. Lu[29], George Bartzokis[29], Ellen Woo[29], Edmond Teng[29], Neill R. Graff-Radford[30], Francine Parfitt[30], Kim Poki-Walker[30], Martin R. Farlow[31], Ann Marie Hake[31], Brandy R. Matthews[31], Jared R. Brosch[31], Scott Herring[31], Christopher H. van Dyck[32], Adam P. Mecca[32], Susan P. Good[32], Martha G. MacAvoy[32], Richard E. Carson[32], Pradeep Varma[32], Howard Chertkow[33], Susan Vaitekunis[33], Chris Hosein[33], Sandra Black[34], Bojana Stefanovic[34], Chris Chinthaka Heyn[34], Ging-Yuek Robin Hsiung[35], Ellen Kim[35], Benita Mudge[35], Vesna Sossi[35], Howard Feldman[35], Michele Assaly[35], Elizabeth Finger[36], Stephen Pasternak[36], Irina Rachinsky[36], Andrew Kertesz[36], Dick Drost[36], John Rogers[36], Ian Grant[37], Brittanie Muse[37], Emily Rogalski[37],

Jordan Robson M.-Marsel Mesulam[37], Diana Kerwin[37], Chuang-Kuo Wu[37], Nancy Johnson[37], Kristine Lipowski[37], Sandra Weintraub[37], Borna Bonakdarpour[37], Nunzio Pomara[38], Raymundo Hernando[38], Antero Sarrael[38], Howard J. Rosen[39], Scott Mackin[39], Craig Nelson[39], David Bickford[39], Yiu Ho Au[39], Kelly Scherer[39], Daniel Catalinotto[39], Samuel Stark[39], Elise Ong[39], Dariella Fernandez[39], Bruce L. Miller[39], Howard Rosen[39], David Perry[39], Raymond Scott Turner[40], Kathleen Johnson[40], Brigid Reynolds[40], Kelly MCCann[40], Jessica Poe[40], Reisa A. Sperling[41], Keith A. Johnson[41], Gad A. Marshall[41], Jerome Yesavage[42], Joy L. Taylor[42], Steven Chao[42], Jaila Coleman[42], Jessica D. White[42], Barton Lane[42], Allyson Rosen[42], Jared Tinklenberg[42], Christine M. Belden[43], Alireza Atri[43], Bryan M. Spann[43], Kelly A. Clark Edward Zamrini[43], Marwan Sabbagh[43], Ronald Killiany[44], Robert Stern[44], Jesse Mez[44], Neil Kowall[44], Andrew E. Budson[44], Thomas O. Obisesan[45], Oyonumo E. Ntekim[45], Saba Wolday[45], Javed I. Khan[45], Evaristus Nwulia[45], Sheeba Nadarajah[45], Alan Lerner[46], Paula Ogrocki[46], Curtis Tatsuoka[46], Parianne Fatica[46], Evan Fletcher[47], Pauline Maillard[47], John Olichney[47], Charles DeCarli[47], Owen Carmichael[47], Vernice Bates[48], Horacio Capote[48], Michelle Rainka[48], Michael Borrie[49], T.-Y Lee[49], Rob Bartha[49], Sterling Johnson[50], Sanjay Asthana[50], Cynthia M. Carlsson[50], Allison Perrin[51], Anna Burke[51], Douglas W. Scharre[52], Maria Kataki[52], Rawan Tarawneh[52], Brendan Kelley[52], David Hart[53], Earl A. Zimmerman[53], Dzintra Celmins[53], Delwyn D. Miller[54], Laura L. Boles Ponto[54], Karen Ekstam Smith[54], Hristina Koleva[54], Hyungsub Shim[54], Ki Won Nam[54], Susan K. Schultz[54], Jeff D. Williamson[55], Suzanne Craft[55], Jo Cleveland[55], Mia Yang[55], Kaycee M. Sink[55], Brian R. Ott[56], Jonathan Drake[56], Geoffrey Tremont[56], Lori A. Daiello[56], Jonathan D. Drake[56], Marwan Sabbagh[57], Aaron Ritter[57], Charles Bernick[57], Donna Munic[57], Akiva Mintz[57], Abigail O'Connelll[58], Jacobo Mintzer[58], Arthur Wiliams[58], Joseph Masdeu[59], Jiong Shi[60], Angelica Garcia[60], Marwan Sabbagh[60], Paul Newhouse[61], Steven Potkin[62], Stephen Salloway[63], Paul Malloy[63], Stephen Correia[63], Smita Kittur[64], Godfrey D. Pearlson[65], Karen Blank[65], Karen Anderson[65], Laura A. Flashman[66], Marc Seltzer[66], Mary L. Hynes[66], Robert B. Santulli[66], Norman Relkin[67], Gloria Chiang[67], Michael Lin[67], Lisa Ravdin[67], Athena Lee[67], Carl Sadowsky[68], Walter Martinez[68], Teresa Villena[68], Elaine R. Peskind[69], Eric C. Petrie[69] & Gail Li[69]

[5]Oregon Health & Science University, Portland, OR, USA. [6]University of Southern California, Los Angeles, CA, USA. [7]University of California, San Diego, CA, USA. [8]University of Michigan, Ann Arbor, MI, USA. [9]Mayo Clinic, Rochester, NY, USA. [10]Baylor College of Medicine, Houston, TX, USA. [11]Columbia University Medical Center, New York, NY, USA. [12]Washington University, St. Louis, MO, USA. [13]University of Alabama at Birmingham, Birmingham, AL, USA. [14]Mount Sinai School of Medicine, New York, NY, USA. [15]Rush University Medical Center, Chicago, IL, USA. [16]Wien Center, Wien, Austria. [17]Johns Hopkins University, Baltimore, MD, USA. [18]University of South Florida, USF Health Byrd Alzheimer's Institute, Tampa, FL, USA. [19]New York University, New York, NY, USA. [20]Duke University Medical Center, Durham, NC, USA. [21]University of Pennsylvania, Philadelphia, PA, USA. [22]University of Kentucky, Lexington, KY, USA. [23]University of Pittsburgh, Pittsburgh, PA, USA. [24]University of Rochester Medical Center, Rochester, NY, USA. [25]University of California Irvine IMIND, Irvine, CA, USA. [26]University of Texas Southwestern Medical School, Dallas, TX, USA. [27]Emory University, Atlanta, GA, USA. [28]University of Kansas, Medical Center, Kansas City, KS, USA. [29]University of California, Los Angeles, CA, USA. [30]Mayo Clinic, Jacksonville, FL, USA. [31]Indiana University, Bloomington, IL, USA. [32]Yale University School of Medicine, New Haven, CT, USA. [33]McGill University, Montreal-Jewish General Hospital, Montreal, QC, Canada. [34]Sunnybrook Health Sciences, Ontario, ON, Canada. [35]U.B.C. Clinic for AD & Related Disorders, Vancouver, BC, Canada. [36]St. Joseph's Health Care, Hamilton, ON, Canada. [37]Northwestern University, Evanston, IL, USA. [38]Nathan Kline Institute, New York, NY, USA. [39]University of California, San Francisco, CA, USA. [40]Georgetown University Medical Center, Georgetown, DC, USA. [41]Brigham and Women's Hospital, Boston, MA, USA. [42]Stanford University, Stanford, CA, USA. [43]Banner Sun Health Research Institute, Sun City, AZ, USA. [44]Boston University, Boston, MA, USA. [45]Howard University, Washington, WA, USA. [46]Case Western Reserve University, Cleveland, OH, USA. [47]University of California, Sacramento, CA, USA. [48]Dent Neurologic Institute, Amherst, MA, USA. [49]Parkwood Institute, London, ON, Canada. [50]University of Wisconsin, Madison, WI, USA. [51]Banner Alzheimer's Institute, Phoenix, AZ, USA. [52]Ohio State University, Columbus, OH, USA. [53]Albany Medical College, Albany, NY, USA. [54]University of Iowa College of Medicine, Iowa City, IA, USA. [55]Wake Forest University Health Sciences, Winston-Salem, NC, USA. [56]Rhode Island Hospital, Providence, RI, USA. [57]Cleveland Clinic Lou Ruvo Center for Brain Health, Las Vegas, NV, USA. [58]Roper St. Francis Healthcare, Charleston, SC, USA. [59]Houston Methodist Neurological Institute, Houston, TX, USA. [60]Barrow Neurological Institute, Phoenix, AZ, USA. [61]Vanderbilt University Medical Center, Nashville, TN, USA. [62]Long Beach VA, Long Beach, CA, USA. [63]Butler Hospital Memory and Aging Program, Butler Hospital, Providence, RI, USA. [64]Neurological Care of CNY, Syracuse, NY, USA. [65]Hartford Hospital, Olin Neuropsychiatry Research Center, Hartford, CT, USA. [66]Dartmouth-Hitchcock Medical Center, Lebanon, PA, USA. [67]Cornell University, Ithaca, NY, USA. [68]Premiere Research Institute, Palm Beach Neurology, Palm Beach, FL, USA. [69]University of Washington, Washington, WA, USA.

