## [Peer Review File · Communications Medicine]

Reviewers' comments:

Reviewer #1 (Remarks to the Author):

The authors present a carefully designed and well-validated ML tool that addresses the problem of early detection of prediagnostic Alzheimer's Disease (AD) using structural neuroimaging. Specifically, they use structural MRI (3D MPRAGE—the standard reference structural scan currently collected in most neuroimaging studies) for training of a Bayesian deep learning neural network which generates an AD score representing the estimated probability of developing AD. The model is trained off structural MRI from the AD Neuroimaging Initiative (ADNI) dataset (healthy controls and AD subjects) and then assessed using the more heterogeneous (in quality of both imaging and additional data) real-world National Alzheimer's Coordinating Center (NACC) dataset. Analysis of the model performance on NACC data showed correlation between a high AD score (over 0.5) and cognitive scores. The model was then applied to a healthy population cohort from the UK Biobank to identify a cohort at risk for AD. Model results from the biobank cohort suggest that those with a higher AD score demonstrate poorer cognitive performance, poorer general health, and the authors identify hypertension and smoking as possible modifiable risk factors with this cohort.

This is very good work and an excellent paper. In addition to the base training on the ADNI dataset, the examination of the generalisability of the model to a challenging out-of-distribution dataset like the NACC is a strength, as are the subsequent application to the UK biobank neuroimaging cohort and the design of confound correction (all the confound corrections on the input data were performed on the ADNI dataset alone, and correctional statistics then applied to the non-training datasets---NACC and biobank). I concur with the authors' assertion that their model is particularly well-validated compared to other models that attempt to produce an AD neuroimaging phenotype.

The authors assert that their approach has improved on previous efforts by identifying individuals with possible early sporadic AD in the UK biobank neuroimaging cohort. They support this assertion by finding a cognitive profile in keeping with AD in those individuals they identified with a high AD score (memory and fluid intelligence in the biobank cohort). They further identified a strong correlation between their model's AD score and a host of cognitive tests performed on subjects in the NACC study (which was not used for training of the model), as well as memory and semantic fluency. These are important and significant results.

One critique of the paper: The ADNI training data used consists entirely of healthy controls and those with diagnosed AD. Ideally, if the model is intended to identify pre-diagnostic AD, the training data would be drawn from structural neuroimages acquired pre diagnosis. While I recognise that a cohort of sufficient size (and uniformity of imaging acquisition and quality) to train a model such as that developed by the authors does not yet exist, I suggest that some discussion of this limitation be included. This doesn't diminish my enthusiasm for the results—the authors have used the available data to very good effect against their purported goal of identifying a cohort of potentially presymptomatic sporadic AD subjects. The results are very promising and I believe will be of high general interest.

Some minor suggestions and comments follow:

- 114: “tanh as the non-linear activation function to leverage both the positive and negative value ranges 115 of the input.” Please provide a justifying reference.
- 136: To assess evidence for group differences, the authors utilize the Region of Practical Equivalence (ROPE). “The ROPE is either set by knowledge of the variable, or set to be 0.1 of the standard deviation of the control group.” Which was used in this case (knowledge of the variable or 0.1 of the std dev of the control group)? Reference?
- 138: Please provide a reference for use of Estimated Log Pointwise Predicted Density (ELPD) as measure of model fit.
- Figure 2: Please expand OD (other degenerative) and OND (other non-degenerative for clarity).
- 167: “Skewed Gaussian families were used for MMSE and CDR Sum of Boxes, otherwise Gaussian distributions were assumed with cauchy distribution priors in all cases.” Please provide ref/justification.
- 177-178: While these are well-known cognitive assessments, I still suggest definition and reference on first use. (i.e., Mini-mental State Examination, Montreal Cognitive Assessment, etc.)
- 216: “probably” should be “probability”?
- 315-317: The authors discuss that they don’t know whether the cohort identified as at risk go on to develop AD or dementia (only a small number have already). They indicate that only 17 in the cohort had diagnosed AD (6 self-reported at the baseline visit). Please put these numbers in context. Of the 17 were any identified as at risk with the author’s model? Were the 6 self-reported at the baseline visit all identified as at risk by the model, or excluded from the analysis?

Reviewer #2 (Remarks to the Author):

This is a very interesting paper showing the use of Bayesian machine learning neural network methods to identify individuals with Alzheimer's Disease. The most interesting part of this work is the effort that this group made on training a model on a certain dataset (ADNI, very specific about AD), validating it on a different dataset (NACC), and then used it to try to identify potential candidates of AD. This work is very interesting and potentially useful. Nevertheless, I see the following problems that the authors would need to address before suggesting the acceptance for publication:

1) This work has clear clinical implications, as they are basically proposing a method (using AI

techniques) for early diagnose of AD. This goal, in itself, is excellent and it aligns very well with the final goals of epidemiological datasets such as UK Biobank. Nevertheless, for a clear acceptance from the clinical community, the proposed method is lacking explainability. As typically happens in ML works, the classifiers are kind of a black box that receives some metrics and return a class with some uncertainty, but does not allow the human user to know how the metrics affected the decisions, which ones (or which combinations of them) are more important, etc. This of course, would not reduce the value of the classifier if the classifier performed a perfect classification. But as it does not, it becomes very important. Clinicians would not accept this kind of classifier easily. I do not expect a neuroimaging ML study to appeal to clinicians, but the lack of explainability should be mentioned in the limitations of the work.

2) The whole "Deep Neural Network Implementation" section lacks a lot of justification. Why 2 hidden layers? Why 128 dimensions? Why hyperbolic tangent function? Why 80% of dropout rate? Why 50 samplings? Why 100 epochs? Why 0.001 learning rate? Why 0.0001 weight decay? Why binary cross entropy loss? If these are "default options" (and I doubt they are, as I can see the parameters and functions hard-coded in main.py, model_statistics.py and models.py), the authors should say so. And reasonably explain why they went for that. If they are not default options, this hyperparameters optimisation should also be justified and the whole process, explained. The authors could have tried different hyperparameters configurations until reaching 74% of accuracy in NACC (This would be considered overfitting by leakage and, if done on purpose, malpractice. A good explanation of all this can be read in Lemm et al paper on NeuroImage: "Introduction to machine learning for brain imaging"). I doubt they did it (74% is not such a good result, after all), but this sections needs way better explanations.

3) It is not clear which subjects of each dataset have been used. Were the 736 subjects from ADNI the full dataset? If not, how were they chosen? Where the 5,209 people from NACC the full dataset? If not, how were they chosen? Were 37,104 subjects from UK Biobank the full dataset? Of what data release (If I am not mistaken, UK Biobank has more than 40,000 datasets right now)? If not, how were they chosen? How were the segmentations assessed for gross abnormalities? Were the ~43,000 datasets visually checked?

4) The whole study is limited by the different number of AD subjects vs controls, as mentioned in the first paragraph of page 5 and the 4th paragraph of page 11. This is explicitly mentioned in the paper, but it would be interesting (in terms of evaluating the real quality of the classification) to see how the results change when the training / validation on ADNI is performed with the same number of AD / controls. This training should be applied to both the whole NACC dataset, and a restricted NACC dataset with the same number of controls and ADs. The actual number of TP, TN, FP, and FN for each of the trainings / validation tests should also be included. The training should also be applied to UK Biobank data. The comment of "desirable property" in page 5 is out of place, as with these diseases, type I errors and type II errors are undesirable in general.

5) Can the authors explain more thoroughly the reasoning behind their peculiar way of controlling for confounds (3rd paragraph, page 3)? At what point is this deconfounding performed? Before the scaling? After the scaling? Why are some features deconfounded

from age, and some other from age, ETIV, and sex? How are these regression models saved and used for other datasets? Are the authors implying that NACC and UK Biobank have been deconfounded using the confounding factors from ADNI? This whole process needs to be better explained, as performing these steps in the wrong order may greatly affect the results by inducing spurious correlations.

6) Could the authors explain a bit better the kind of figures they are showing in Figures 3 and later? It looks like a boxplot, but the difference between the thin and the not-so-thin "horizontal bars" is not very clear. Is the thick bar the inter-quartile range? My doubts come from Figure 5, where the boxplots for the negative group (which has an N that is two orders of magnitude bigger) is narrower than the boxplot for the positive group. I would expect a lot more variability in most of these metrics on the negative group (the bigger group). Otherwise, I am not understanding these figures correctly.

The authors have done a great job. All in all, I liked the paper, although it cannot be accepted as is.

Minor typos:

- Suggest  Suggests (3rd line of the abstract).
- Have  Has (10th line of the abstract).
- An neuroimaging  A neuroimaging (Lines 43/44).
- Have  Has (Line 45).
- Report  Reports (line 47).

After this, I stopped checking the grammar. The paper may benefit from using an automated spelling and grammar check.

Reviewer #3 (Remarks to the Author):

The manuscript report results from the application of a Bayesian machine learning neural network model trained on about ~700 subjects from the ADNI cohort, which was also validated on about 5000 subjects from a real-world dataset of patients with AD and other dementias (National Alzheimer's Coordinating Centre). The model was then applied on about ~17000 healthy controls from the UK Biobank to identify a cohort of subjects at risk of AD (or with a score suggestive to have an AD imaging phenotype). The authors found that the UK Biobank cohort resulting at risk form imaging parameters also had lower scores on measures of fluid intelligence, and -to less extent- other cognitive measures including numeric memory. The idea is extremely interesting. The numbers are huge and show the potential of big-data usage.

However, I am afraid that the main point that the cohort identified as "at risk" may be in the preclinical phase of AD is little supported.

First of all, fluid intelligence is not known to be a cognitive domain affected in the early, preclinical phases of AD. Which test of fluid intelligence was used in the UK Biobank? Beyond fluid intelligence, there was also some effect on tests of numeric memory, and it is not clear from the manuscript how this was measured either. There is also mention of a difference on

“reaction time for correct trials” but it is not specified to which cognitive test these trials are referring to: you can have reaction times for correct trials in a number of cognitive tests, measuring a number of different cognitive domains. These points are not trivial as the authors suggest that the 1,304 UK Biobank subjects identified as “at risk” may be in the preclinical phase of AD, therefore it would be absolutely important to characterise them better.

Following on this, I would like to know what is the age of the 1,304 subjects at risk and whether it is different from the age of the 36,663 subjects not at risk. Indeed, some of the non-cognitive measures such as frailty, poor health and reported falls are suggestive that there might be a strong effect of age. I understand that age was included during the training phase of the model in the ADNI cohort (page 3). Would it be possible to have a measure of the “importance” of age in the obtained AD score? Would it also be possible to have a ranking of the importance of the 155 brain features included in the model?

In the methods paragraph it is rightly stated that the UK Biobank is a non-clinical cohort. This is very important as the whole point of the study is to identify subjects at risk when they are still cognitively healthy. However, in the discussion it is stated that 6 subjects had self-reported dementia at the baseline visit: My understanding is that if the study is about identifying preclinical AD, these subjects should have been excluded. Similarly, it would be helpful to know if hippocampal volumetry alone would have identified the subjects at risk, or how much the proposed model improves identification above known risk factors such as age and hippocampal volume alone.

Minor issues:

1) The authors state in the abstract, introduction and discussion that structural imaging studies on preclinical AD have been mainly done on genetic forms of AD, whereas it is unclear if structural changes can be detected in sporadic AD. This is not correct as the literature on structural imaging changes in the preclinical phase of sporadic AD is vast and goes back of 10-15 years: please see, as an example, Tondelli et al 2012, den Heijer et al. 2006, Smith CD et al 2007 to quote some of the earliest studies.

2) The statement “the role and timing of b-amyloid PET abnormalities remain uncertain in the detection of presymptomatic Alzheimer’s disease” in the discussion is arguable and misleading. Please note that markers of amyloid deposition are now considered biomarkers of preclinical AD (see Sperling et al 2011)

3) I strongly suggest refraining from using the expression “AD positive participants” to indicate subjects who resulted to have a positive AD score: please do keep referring to the “score”. The distinction is crucial, and relates to my major observation above, i.e. the model identifies subjects who might possibly have preclinical AD on the basis of structural imaging, but this is not proven.

4) In the discussion it is stated “key cognitive domains of AD including memory and fluid intelligence”: this is not correct (see above)

Reply to the Reviewers

Re: Manuscript ID COMMSMED-21-0614

“Identifying healthy individuals with Alzheimer neuroimaging phenotypes in the UK Biobank”

We would like to thank the editor and the editorial board for the opportunity to resubmit a revised version of our paper, which has been amended in accordance with the reviewer’s suggestions. We would also like to thank the reviewers for their insightful feedback. We are encouraged by the positive comments about how “very/extremely interesting” this work is, as well as its potential utility (reviewers 2 and 3). Likewise, we greatly appreciate the positive remarks from reviewer 1 highlighting that our experiments were carefully designed and that their comments do not diminish their enthusiasm for the results.

We conducted an in-depth revision of our paper and feel we have addressed all main concerns expressed by the reviewers’ comments in the revised paper. We are confident that this revised version is improved with respect to the previous one, and therefore hope that the reviewers will be available to re-read our work.

All changes in the revised version of the document are highlighted in strikeout red (for removed parts) and underline blue (for added parts). The anonymised repository linked to this paper was updated according to the new experiments which have been conducted for the purpose of this revision.

Reviewer #1, comment #1

The authors present a carefully designed and well-validated ML tool that addresses the problem of early detection of prediagnostic Alzheimer’s Disease (AD) using structural neuroimaging. Specifically, they use structural MRI (3D MPRAGE—the standard reference structural scan currently collected in most neuroimaging studies) for training of a Bayesian deep learning neural network which generates an AD score representing the estimated probability of developing AD. The model is trained off structural MRI from the AD Neuroimaging Initiative (ADNI) dataset (healthy controls and AD subjects) and then assessed using the more heterogeneous (in quality of both imaging and additional data) real-world National Alzheimer’s Coordinating Center (NACC) dataset. Analysis of the model performance on NACC data showed correlation between a high AD score (over 0.5) and cognitive scores. The model was then applied to a healthy population cohort from the UK Biobank to identify a cohort at risk for AD. Model results from the biobank cohort suggest that those with a higher AD score demonstrate poorer cognitive performance, poorer general health, and the authors identify hypertension and smoking as possible modifiable risk factors with this cohort.

This is very good work and an excellent paper. In addition to the base training on the ADNI dataset, the examination of the generalisability of the model to a challenging out-of-distribution dataset like the NACC is a strength, as are the subsequent application to the UK biobank neuroimaging cohort and the design of confound correction (all the confound corrections on the input data were performed on the ADNI dataset alone, and correctional statistics then applied to the non-training datasets—NACC and biobank). I concur with the authors’ assertion that their model is particularly well-validated compared to other models that attempt to produce an AD neuroimaging phenotype.

The authors assert that their approach has improved on previous efforts by identifying individuals with possible early sporadic AD in the UK biobank neuroimaging cohort. They support this assertion by finding a cognitive profile in keeping with AD in those individuals they identified with a high AD score (memory and fluid intelligence in the biobank cohort). They further identified a strong correlation between their model’s AD score and a host of cognitive tests performed on subjects in the NACC study (which was not used for training of the model), as well as memory and semantic fluency. These are important and significant results.

Our response #1.1

We thank the reviewer for the kind comments about our paper.

Reviewer #1, comment #2

One critique of the paper: The ADNI training data used consists entirely of healthy controls and those with diagnosed AD. Ideally, if the model is intended to identify pre-diagnostic AD, the training data would be

drawn from structural neuroimages acquired pre diagnosis. While I recognise that a cohort of sufficient size (and uniformity of imaging acquisition and quality) to train a model such as that developed by the authors does not yet exist, I suggest that some discussion of this limitation be included. This doesn't diminish my enthusiasm for the results—the authors have used the available data to very good effect against their purported goal of identifying a cohort of potentially presymptomatic sporadic AD subjects. The results are very promising and I believe will be of high general interest.

Our response #1.2

We thank the reviewer for highlighting this important point, and we agree that an ideal model would be trained on pre-symptomatic data that is not currently available. We have added the following to the “Discussion” section:

An ideal dataset for training our model would be comprised of people prior to a diagnosis of Alzheimer's disease, which does not yet exist in a sufficient size to train a deep learning model.

Reviewer #1, comment #3

114: “*tanh* as the non-linear activation function to leverage both the positive and negative value ranges 115 of the input.” Please provide a justifying reference.

Our response #1.3

We did not include a reference to the *tanh* function because it is just a standard mathematical function. Similarly to other architectural decisions in the field of machine learning, there is no single, systematic recommendation regarding which non-linear activation function to use. Our choice considered the fact that the input to the neural network included not only positive values (as with most of the applications of neural networks nowadays), but also negative values. Therefore, we decided that it would be sensible to choose a non-linear activation that could leverage this range of values, instead of the commonly used ReLU which ignores negative values. In this sense, we preferred the hyperbolic tangent (i.e. *tanh*) instead of other activations (e.g. ELU, Leaky RELU, PReLU, among others) as this is a symmetric function in which negative inputs will be mapped strongly negative, and positive inputs will be mapped strongly positive. This symmetric property allows for a normalisation of layer's outputs, therefore avoiding using other mechanisms like batch normalisation and allowing our neural network to be less complex.

We changed the corresponding sentence (in the “Deep Neural Network Implementation” subsection) to the following, hoping it illustrates better our thought process, even though we cannot provide a specific reference: As depicted in figure 1, we implemented a neural network with two hidden layers, each with 128 dimensions. (...) We used the hyperbolic tangent function (*tanh*()) as the non-linear activation function to leverage both the positive and negative value ranges of the input. This non-linear activation is a symmetric function in which negative inputs will be mapped strongly negative, and positive inputs will be mapped strongly positive; this symmetric property allows for a normalisation of layer's outputs, therefore avoiding using other mechanisms like batch normalisation and allowing our neural network to be less complex.

Reviewer #1, comment #4

136: To assess evidence for group differences, the authors utilize the Region of Practical Equivalence (ROPE). “The ROPE is either set by knowledge of the variable, or set to be 0.1 of the standard deviation of the control group.” Which was used in this case (knowledge of the variable or 0.1 of the std dev of the control group)? Reference?

Our response #1.4

We thank the reviewer for pointing out this omission. We have provided a reference to John Kruschke's paper that recommends the ROPE of 0.1, named “Rejecting or Accepting Parameter Values in Bayesian Estimation” [8].

Reviewer #1, comment #5

138: Please provide a reference for use of Estimated Log Pointwise Predicted Density (ELPD) as measure of model fit.

Our response #1.5

We thank the reviewer for this suggestion. As a result, we have added Yao et al.'s paper "Using stacking to average bayesian predictive distributions" [16] as a reference to that paragraph.

Reviewer #1, comment #6

Figure 2: Please expand OD (other degenerative) and OND (other non-degenerative for clarity).

Our response #1.6

Thanks for highlighting this potential source of confusion. At the end of the "Datasets" subsection of the "Methods" section, we added the following clarification:

Other degenerative disorders (OD) correspond to NACC labels "Vascular brain injury or vascular dementia including stroke", "Lewy body disease (LBD)", "Prion disease (CJD, other)", "FTLD, other", "Corticobasal degeneration (CBD)", "Progressive supranuclear palsy (PSP)", and "FTLD with motor neuron disease (e.g., ALS)". Other non-degenerative disorders (OND) correspond to NACC labels "Depression", "Other neurologic, genetic, or infectious condition", "Cognitive impairment for other specified reasons (i.e., written-in values)", "Anxiety disorder", "Cognitive impairment due to medications", "Other psychiatric disease", "Cognitive impairment due to systemic disease or medical illness", "Traumatic brain injury (TBI)", "Cognitive impairment due to alcohol abuse", "Bipolar disorder", and "Schizophrenia or other psychosis".

Reviewer #1, comment #7

167: "Skewed Gaussian families were used for MMSE and CDR Sum of Boxes, otherwise Gaussian distributions were assumed with cauchy distribution priors in all cases." Please provide ref/justficiation.

Our response #1.7

We thank the reviewer for the opportunity to further elaborate on this point. The choice of prior distribution was influenced by the known distributions of these tests that tend to be highly skewed because of ceiling effect in healthy people. We changed that sentence (in the subsection "Clinical scores" under "ADNI" subsection) to:

Given the distribution of the observed data, skewed Gaussian families were used for MMSE and CDR Sum of Boxes, otherwise Gaussian distributions were assumed with cauchy distribution priors in all cases.

Reviewer #1, comment #8

177-178: While these are well-known cognitive assessments, I still suggest definition and reference on first use. (i.e., Mini-mental State Examination, Montreal Cognitive Assessment, etc.)

Our response #1.8

We completely agree, this was an oversight on our part and the MMSE, MoCA, CDR sum of Boxes, Trails tests, WAIS, and Boston naming test are now referenced.

Reviewer #1, comment #9

216: "probably" should be "probability"?

Our response #1.9

We thank the reviewer for spotting this typo, we corrected it in the resubmitted version.

Reviewer #1, comment #10

315-317: The authors discuss that they don't know whether the cohort identified as at risk go on to develop AD or dementia (only a small number have already). They indicate that only 17 in the cohort had diagnosed AD (6 self-reported at the baseline visit). Please put these numbers in context. Of the 17 were any identified as at risk with the author's model? Were the 6 self-reported at the baseline visit all identified as at risk by the model, or excluded from the analysis?

Our response #1.10

Thank you for this comment, we should have been clearer that the 6 people who had a diagnosis of dementia were excluded from the analysis in the paper. All of these 6 had positive AD scores (range 0.60 to 0.95). This information has been added to the results section. Of the 11 who converted, 7 had positive AD scores, and a further one was just below the cut-off (0.49).

Reviewer #2, comment #1

This is a very interesting paper showing the use of Bayesian machine learning neural network methods to identify individuals with Alzheimer's Disease. The most interesting part of this work is the effort that this group made on training a model on a certain dataset (ADNI, very specific about AD), validating it on a different dataset (NACC), and then used it to try to identify potential candidates of AD. This work is very interesting and potentially useful. Nevertheless, I see the following problems that the authors would need to address before suggesting the acceptance for publication

Our response #2.1

We sincerely appreciate the reviewer for the interest in the paper and our approach.

Reviewer #2, comment #2

This work has clear clinical implications, as they are basically proposing a method (using AI techniques) for early diagnose of AD. This goal, in itself, is excellent and it aligns very well with the final goals of epidemiological datasets such as UK Biobank. Nevertheless, for a clear acceptance from the clinical community, the proposed method is lacking explainability. As typically happens in ML works, the classifiers are kind of a black box that receives some metrics and return a class with some uncertainty, but does not allow the human user to know how the metrics affected the decisions, which ones (or which combinations of them) are more important, etc. This of course, would not reduce the value of the classifier if the classifier performed a perfect classification. But as it does not, it becomes very important. Clinicians would not accept this kind of classifier easily. I do not expect a neuroimaging ML study to appeal to clinicians, but the lack of explainability should be mentioned in the limitations of the work.

Our response #2.2

We would like to clarify that we do not consider this model to be sufficient for making a confident diagnosis in a clinical context. However, it may be helpful in screening to select a high risk population suitable for prevention/disease modifying treatment trials. In addition, we deliberately chose to use a relatively 'shallow' neural network. There is increasing work that such networks can be interrogated.

We agree that explainability and transparency are important parts of the clinical translation of AI tools. We therefore added additional analysis interrogating the neural network, adding the following extra section in supplementary material named "Model explainability" (Figures R1 and R2 correspond to figures S4 and S5 in supplementary material):

We investigated the potential explainability of our model using SHapley Additive exPlanations (SHAP), a unified framework for interpreting predictions.

Figure R1 shows the aggregated feature impact on the model output in the ADNI validation set. In the figure, a point represents a sample from the dataset and its colour is the value of that feature rather than the importance on the model output. The y-axis contains the 20 most important input features, ranked by the aggregated magnitude of impact on the model output across all the samples (the 20th row is an aggregation

of the contribution of all the remaining 136 features after the 19 most important). Each feature is assigned a SHAP value (in the x-axis) which represents the marginal impact (i.e., importance) on model output or, in other words, both the magnitude and direction of the feature's contribution. A higher SHAP value means that that feature contributed towards a higher predicted value in the model's output.

It is possible to interpret the contributions of individual brain regions to the model predictions. For instance, the cortical thickness of the left hemisphere's entorhinal area has an almost inverse effect in the model output: a lower value of this feature drives up Alzheimer's disease prediction with a similar magnitude as when a higher value of this feature drives the prediction down. This effect can be seen, as expected, in almost all the important features, with some exceptions (e.g., the cortical thicknesses of the right hemisphere's transverse temporal area, and the volume of the left hemisphere's precentral area).

Figure R1: **Contribution of the most important features across the ADNI validation set.** For each feature represented in each row, vertical dispersion stands for the data points which share the same SHAP value for that feature. Each feature value is colour-coded from the highest (i.e. red) to the lowest value (i.e. blue). Higher SHAP values, which are distinct from the actual feature values, mean they contribute in a positive direction to the final predicted variable.

Besides allowing the interpretation and analysis of output drivers on an aggregated (i.e. global) level, SHAP also enables the analysis of individuals. As a reminder, SHAP values represent the change in the expected model prediction conditioned on each feature, therefore explaining the contribution of that feature towards the difference between the average model prediction and the actual final prediction. In Figure R2 we show an example of the most important features driving the AD score in two individuals with a high AD score (0.957 in the top pane) and a low AD score (0.005 in the bottom pane). These plots decompose the drivers of predictions for one single sample each. The y-axis contains the most important features driving the prediction and the corresponding raw value in lighter grey, and the x-axis contains the SHAP value corresponding to the impact on final prediction from the baseline prediction across the population (represented by $E[f(X)]$). The SHAP value of each individual feature is detailed in the arrows that move the prediction from the $E[f(X)]$ baseline. A striking difference between the two plots is that for the top one, the most important features drive most of the output value, but in the sample on the bottom, the remaining 146 other features (in total) have a much greater effect. This could point an expert to a more wider analysis on the whole brain (in the bottom case), while the analysis on the top case can possibly be more focused on a handful of brain regions.

Figure R2: **Contribution of the most important features in two samples of the ADNI validation set.** The most important features driving different final outputs in two distinct people with a high (above) and low (below) AD score.

We added the following sentence to the end of “Model Evaluation and Performance” subsection under “Results” to reference these results: In supplementary figures S4 and S5, we illustrate possible explainability capacities of our model when using it together with SHapley Additive exPlanations (SHAP) [61], a unified framework for interpreting predictions.

Reviewer #2, comment #3

The whole “Deep Neural Network Implementation” section lacks a lot of justification. Why 2 hidden layers? Why 128 dimensions? Why hyperbolic tangent function? Why 80% of dropout rate? Why 50 samplings? Why 100 epochs? Why 0.001 learning rate? Why 0.0001 weight decay? Why binary cross entropy loss? If these are “default options” (and I doubt they are, as I can see the parameters and functions hard-coded in main.py, model_statistics.py and models.py), the authors should say so. And reasonably explain why they went for that. If they are not default options, this hyperparameters optimisation should also be justified and the whole process, explained. The authors could have tried different hyperparameters configurations until reaching 74% of accuracy in NACC (This would be considered overfitting by leakage and, if done on purpose, malpractice. A good explanation of all this can be read in Lemm et al paper on NeuroImage: “Introduction to machine learning for brain imaging”). I doubt they did it (74% is not such a good result, after all), but

this sections needs way better explanations.

Our response #2.3

Thanks for providing us with the opportunity to clarify our modelling choices. We deliberately chose not to train the model on the NACC dataset after much discussion, and agreed that to do so to improve accuracy in the NACC would invalidate the dataset’s role for completely independent validation. We added the following paragraph at the end of the “Datasets” subsection of the “Methods” section with our justification:

We deliberately chose not to train the model on the NACC dataset. The argument to train on this dataset rather than ADNI would be the larger dataset available, but the NACC dataset is much more ‘noisy’ in the sense that the diagnostic labels are clinical rather than biomarker supported (as in ADNI). It is highly likely that if we had trained in the NACC data and validated in the ADNI dataset our results would have looked better in terms of raw accuracy, but we consider that this would be falsely reassuring given the highly selected nature of the ADNI cohort.

Given the very specific task and data used, we had to design our own ML model; it would not make sense to use convolutional neural networks, recurrent neural networks, or other more complex architectures with our type of input (i.e., 1D flatten features). In this sense, we used fully connected layers (sometimes referred as “dense” layers), for which there is not a “portfolio” of previously studied models in the literature like there is for other networks (eg., CNNs). We want to highlight that we checked for convergence of our neural network with ADNI training/validation splits to be sure that the choice of hyperparameters was sensible, and publicly shared all our training logs¹ for transparency and further reproducibility by the community. Similarly to other architectural decisions in the field of machine learning, there is no single, systematic recommendation regarding which choices of hyperparameters to use, so we expanded the explanations in “Deep Neural Network Implementation” (under “Methods” section) to the following sentences, hoping these points are clearer:

As depicted in figure 1, we implemented a neural network with two hidden layers, each with 128 dimensions. Given the small dataset size, we empirically found these hyperparameters to give stable learning curves, thus avoiding a deeper neural network which could more easily overfit on the small training data. Dropout layers were added after each hidden layer with a high dropout rate (i.e., 80%) to help in avoiding overfitting given the small neural network size; a smaller dropout rate was empirically found to provide slightly worse metrics on the ADNI validation set. We used the hyperbolic tangent function ($\tanh()$) as the non-linear activation function to leverage both the positive and negative value ranges of the input. This non-linear activation is a symmetric function in which negative inputs will be mapped strongly negative, and positive inputs will be mapped strongly positive; this symmetric property allows for a normalisation of layer’s outputs, therefore avoiding using other mechanisms like batch normalisation and allowing our neural network to be less complex. The *sigmoid* function was applied to the last output node to give a value between 0 and 1 to represent the likelihood that the individual has Alzheimer’s disease.

Monte Carlo dropout was employed by sampling (i.e. making a forward pass) 50 times from the model, after which a mean and standard deviation was calculated. The mean corresponds to the final model prediction (i.e. likelihood of Alzheimer’s disease), and the standard deviation represents the uncertainty of the model. A higher number of samples would bring an increased statistical power to the Bayesian approximation process, but it would also increase the inference time, thus this number (i.e., 50) was chosen as a good compromise in accordance with previous literature [4].

We highlight that a more systematic hyperparameter search could potentially bring better metrics on the ADNI validation set, but we consider such extensive exploration to be beyond the scope of this work, and with diminished returns given the small size of the ADNI dataset.

(...)

The model was implemented using Pytorch [10] and trained for 100 epochs using the Adam optimiser [7] with the default learning rate of 0.001. Training convergence was achieved under 50 epochs, therefore 100 epochs for training was considered reasonable. A small weight decay was set to 0.0001 to help with regularisation, and binary cross entropy loss was chosen given the prediction of a binary output.

¹In our public repository we point to a *Weights & Biases* log, which we do not include here as that would break the double-blind review process. However, if the reviewer prefers, we could send anonymised data points through the journal’s editor.

Reviewer #2, comment #4

It is not clear which subjects of each dataset have been used. Were the 736 subjects from ADNI the full dataset? If not, how were they chosen? Where the 5,209 people from NACC the full dataset? If not, how were they chosen? Were 37,104 subjects from UK Biobank the full dataset? Of what data release (If I am not mistaken, UK Biobank has more than 40,000 datasets right now)? If not, how were they chosen? How were the segmentations assessed for gross abnormalities? Were the 43,000 datasets visually checked?

Our response #2.4

The fully available datasets at the time of the analysis were used in all cases. We have particularly clarified in the method section that all ADNI studies were included. It is true that more neuroimaging data has been more recently made available by the UK Biobank, but this happened after the analysis took place. Given the size of the database, visual checks were not performed.

Reviewer #2, comment #5

The whole study is limited by the different number of AD subjects vs controls, as mentioned in the first paragraph of page 5 and the 4th paragraph of page 11. This is explicitly mentioned in the paper, but it would be interesting (in terms of evaluating the real quality of the classification) to see how the results change when the training / validation on ADNI is performed with the same number of AD / controls. This training should be applied to both the whole NACC dataset, and a restricted NACC dataset with the same number of controls and ADs. The actual number of TP, TN, FP, and FN for each of the trainings / validation tests should also be included.

Our response #2.5

We thank the reviewer for bringing interesting possible directions for our work. With regards to train the model on the NACC dataset, we believe we have given a reasonable justification against this idea in our response # 2.3.

With regards to the actual number of TP/TN/FP/FN, we added those to figure 2's caption.

As a result of the reviewer's comment, we conducted an experiment with a balanced training ADNI dataset for comparison. We added the following extra section in supplementary material named "Training on a balanced ADNI training set" (Table R1 corresponds to table S8 in supplementary material):

We employed the same training pipeline (i.e., same preprocessing steps and hyperparameters) with a balanced training ADNI set (i.e., by removing 60 control people to have 301 people both in the AD and control groups), and evaluated this new model on the ADNI and NACC validation sets. Resulting metrics can be seen on table S8. Overall, metrics are very similar or slightly worse when compared to the model trained on the unbalanced dataset (with the exception of the sensitivity metric), which is expected as we are reducing the number of training samples on an already small dataset.

We added the following sentence to the end of "Model Evaluation and Performance" subsection under "Results" to reference these results:

For comparison purposes, we trained the model with the same hyperparameters and preprocessing steps on a balanced training ADNI set (i.e., by having the same number of people in both the AD and control groups). In general, evaluation metrics do not improve on the ADNI and NACC validation sets with this setting, which is expected given the already small training set size (see supplementary table S8).

Reviewer #2, comment #6

The training should also be applied to UK Biobank data.

Our response #2.6

Although we agree with the reviewer that it might be interesting to use the UK Biobank for training, that would not be possible; this dataset is mostly constituted by healthy individuals and there are not enough people with Alzheimer's disease to train a classification model. Instead, the UK Biobank dataset is

Table R1: Performance metrics across datasets with a model trained on a balanced ADNI training set (i.e., same number of people with AD diagnosis and controls), using a cut-off of and AD score of 0.5 and employing inference using MC Dropout with 50 samples. AUC=Area under the ROC curve. PPV=Positive predictive value. NPV=Negative predictive value. Results with the unbalanced (i.e., original) dataset presented for comparison.

Dataset	Accuracy	AUC	Sensitivity	Specificity	PPV/Precision	NPV
ADNI test set [unbalanced]	0.92	0.97	0.90	0.93	0.90	0.93
ADNI test set [balanced]	0.89	0.97	0.87	0.91	0.87	0.91
NACC (only AD/Control) [unbalanced]	0.74	0.79	0.68	0.78	0.65	0.80
NACC (only AD/Control) [balanced]	0.74	0.79	0.7	0.76	0.64	0.81
NACC (AD/All) [unbalanced]	0.72	0.76	0.68	0.73	0.56	0.83
NACC (AD/All) [balanced]	0.71	0.76	0.7	0.72	0.55	0.83

appropriately independent dataset for our purpose of finding healthy people with a high risk of developing dementia in a healthy cohort and no known diagnosis of dementia.

Reviewer #2, comment #7

The comment of "desirable property" in page 5 is out of place, as with these diseases, type I errors and type II errors are undesirable in general.

Our response #2.7

We agree that labelling the bias a 'desirable property' is poorly worded, and have updated the corresponding sentence (under "Results"'s "Model Evaluation and Performance" subsection) to the following:

The bias towards a relatively high negative predictive value is relatively better than the reverse situation given the application to UK Biobank data where the rate of Alzheimer's disease will be substantially lower than either ADNI or NACC, so there is a greater risk of misclassifying healthy people as having Alzheimer's disease.

Reviewer #2, comment #8

Can the authors explain more thoroughly the reasoning behind their peculiar way of controlling for confounds (3rd paragraph, page 3)? At what point is this deconfounding performed? Before the scaling? After the scaling? Why are some features deconfounded from age, and some other from age, ETIV, and sex? How are these regression models saved and used for other datasets? Are the authors implying that NACC and UK Biobank have been deconfounded using the confounding factors from ADNI? This whole process needs to be better explained, as performing these steps in the wrong order may greatly affect the results by inducing spurious correlations.

Our response #2.8

We apologise for the possible lack of clarity and sincerely hope that this answer can clear up the reviewer's doubts. There is a well established association between volume, estimated TIV and sex, but this association does not exist for cortical thickness values.

We extended the corresponding paragraph in the "ADNI Preprocessing" subsection (as well as added documentation to the public repository), which we believe should have clarified all the questions about our preprocessing steps:

To regress out confounds, 155 distinct linear regression models (one for each input feature) were fitted to the training set using ordinary least squares (OLS) implemented in *statsmodels* [12]. For each of the 68 cortical thickness features, the independent variable to be regressed out was age. For the remaining 87 volume features, the independent variables were age, estimated total intracranial volume, and sex. These 155 regression models (as defined using the *statsmodels* package) were saved in disk to be later employed

on the ADNI validation set, NACC, and UK Biobank datasets. We ensure no data leakage in the training and evaluation processes by deconfounding the validation/test sets (i.e., NACC, UK Biobank, and ADNI validation set) using only ADNI confounding factors learned from the training set.

After the data is deconfounded, each 155 input feature was separately scaled to zero mean and unit variance for numerical stability when training a neural network using *Scikit-learn* [11]. Normalisation statistics were once again calculated only using the ADNI training set. Values in the validation/test sets (i.e., NACC, UK Biobank, and ADNI validation set) were normalised using the normalisation statistics from the ADNI training set, after the deconfound process.

Reviewer #2, comment #9

Could the authors explain a bit better the kind of figures they are showing in Figures 3 and later? It looks like a boxplot, but the difference between the thin and the not-so-thin "horizontal bars" is not very clear. Is the thick bar the inter-quartile range? My doubts come from Figure 5, where the boxplots for the negative group (which has an N that is two orders of magnitude bigger) is narrower than the boxplot for the positive group. I would expect a lot more variability in most of these metrics on the negative group (the bigger group). Otherwise, I am not understanding these figures correctly.

Our response #2.9

These plots show the posterior distribution of the Bayesian analysis, estimating the mean. The thick and thin lines represent the credible intervals, not the distribution of the observed data. We have amended figure 3's caption to make this more clear:

These distributions represent the Bayesian posterior estimates of the mean of cognitive tests in AD score positive and negative groups, with the Region Of Practical Equivalence (ROPE) as a shaded column.

Reviewer #2, comment #10

The authors have done a great job. All in all, I liked the paper, although it cannot be accepted as is.

Minor typos:

- Suggest --i>Suggests (3rd line of the abstract).
- Have Has (10th line of the abstract).
- An neuroimaging A neuroimaging (Lines 43/44).
- Have Has (Line 45).
- Report Reports (line 47).

After this, I stopped checking the grammar. The paper may benefit from using an automated spelling and grammar check.

Our response #2.10

We have spotted and corrected a few typos in the revised version of the paper. Thank you for highlighting this issue and spotting a few typos for us.

Reviewer #3, comment #1

The manuscript report results from the application of a Bayesian machine learning neural network model trained on about 700 subjects from the ADNI cohort, which was also validated on about 5000 subjects from a real-world dataset of patients with AD and other dementias (National Alzheimer's Coordinating Centre). The model was then applied on about ~17000 healthy controls from the UK Biobank to identify a cohort of subjects at risk of AD (or with a score suggestive to have an AD imaging phenotype). The authors found that the UK Biobank cohort resulting at risk form imaging parameters also had lower scores on measures of fluid intelligence, and -to less extent- other cognitive measures including numeric memory. The idea is extremely interesting. The numbers are huge and show the potential of big-data usage.

Our response #3.1

We greatly welcome the reviewer's positive comments on our work and appreciate the recognition over our

efforts. We also hope that future investigations could extend on our work and have fruitful impacts in the field.

Reviewer #3, comment #2

However, I am afraid that the main point that the cohort identified as “at risk” may be in the preclinical phase of AD is little supported.

First of all, fluid intelligence is not known to be a cognitive domain affected in the early, preclinical phases of AD. Which test of fluid intelligence was used in the UK Biobank? Beyond fluid intelligence, there was also some effect on tests of numeric memory, and it is not clear from the manuscript how this was measured either.

Our response #3.2

The fluid intelligence test used in the UK Biobank assessment is a bespoke set of 13 questions administered via a touchscreen. We accept that the tests used are not previously validated and differ from those typically used to assess fluid intelligence. However, the validity of these tests has been assessed separately.

To make these points clearer, we added the following paragraph to the “Datasets” subsection (under the “Methods” section):

Some cognitive tests used in the UK Biobank are non-standard including fluid intelligence², numeric memory³, matrix reasoning⁴, and reaction time⁵. The validity of these tests has been assessed separately, finding moderate to high validity for the cognitive tests used [3].

A very recent paper has examined those in the UK Biobank who converted to Alzheimer’s disease over the course of the UK Biobank [14]; the authors have found that fluid intelligence was impaired early in Alzheimer’s disease, years prior to diagnosis. This is in accordance with findings in Mild Cognitive impairment [1] and presymptomatic Alzheimer’s disease [5].

Reviewer #3, comment #3

There is also mention of a difference on “reaction time for correct trials” but it is not specified to which cognitive test these trials are referring to: you can have reaction times for correct trials in a number of cognitive tests, measuring a number of different cognitive domains. These points are not trivial as the authors suggest that the 1,304 UK Biobank subjects identified as “at risk” may be in the preclinical phase of AD, therefore it would be absolutely important to characterise them better.

Our response #3.3

We agree with the reviewer that we should have been more specific about the tasks used in the UK Biobank. The reaction time task used in the UK Biobank is a distinct task, based on the card game ‘snap’. The reference for this “reaction time” test was added together with the changes highlighted in our previous response # 3.2 which were added to the “Methods” section.

Reviewer #3, comment #4

Following on this, I would like to know what is the age of the 1,304 subjects at risk and whether it is different from the age of the 36,663 subjects not at risk. Indeed, some of the non-cognitive measures such as frailty, poor health and reported falls are suggestive that there might be a strong effect of age. I understand that age was included during the training phase of the model in the ADNI cohort (page 3).

Our response #3.4

This is an important point, and we thank the reviewer for raising it. Perhaps surprisingly, there was very little difference in age between the AD score positive and negative groups. We have added a line to the

²<https://biobank.ctsu.ox.ac.uk/crystal/ukb/docs/Fluidintelligence.pdf>

³https://biobank.ctsu.ox.ac.uk/crystal/ukb/docs/numeric_memory.pdf

⁴<https://biobank.ctsu.ox.ac.uk/crystal/label.cgi?id=501>

⁵<https://biobank.ctsu.ox.ac.uk/crystal/label.cgi?id=100032>

results:

The group with a positive AD score were only slightly older than the AD score negative group (1.79 years, CI 1.39 to 2.21).

Reviewer #3, comment #5

Would it be possible to have a measure of the “importance” of age in the obtained AD score? Would it also be possible to have a ranking of the importance of the 155 brain features included in the model?

Our response #3.5

Age was not used to generate the AD score, but the effect of age was regressed out during the preprocessing phase.

Reviewer #3, comment #6

In the methods paragraph it is rightly stated that the UK Biobank is a non-clinical cohort. This is very important as the whole point of the study is to identify subjects at risk when they are still cognitively healthy. However, in the discussion it is stated that 6 subjects had self-reported dementia at the baseline visit: My understanding is that if the study is about identifying preclinical AD, these subjects should have been excluded. Similarly, it would be helpful to know if hippocampal volumetry alone would have identified the subjects at risk, or how much the proposed model improves identification above known risk factors such as age and hippocampal volume alone.

Our response #3.6

We have clarified in the paper that the 6 people reporting a diagnosis of dementia at baseline were excluded from the main analysis.

We compared the prediction power of hippocampal atrophy alone in the ADNI cohort, and it was less predictive. We added a new section to the supplementary material named “Predictive power of hippocampal volume” with the following information (Table R2 corresponds to table S7 in supplementary material):

We fitted a linear regression model to the training (ADNI) set using ordinary least squares (OLS), in which the dependent variable was AD diagnosis, and independent variables were left hippocampus volume, right hippocampus volume, age, estimated total intracranial volume, and sex. This model was then employed on the ADNI test set, and resulting metrics can be seen in table S7.

Table R2: Performance metrics evaluated on ADNI test set using the main model presented in the paper, and a linear regression model based on hippocampal volume. AUC=Area under the ROC curve. PPV=Positive predictive value. NPV=Negative predictive value.

Model	Accuracy	AUC	Sensitivity	Specificity	PPV/Precision	NPV
Deep Learning	0.92	0.97	0.90	0.93	0.90	0.93
Linear regression	0.80	0.80	0.77	0.82	0.74	0.84

We added the following sentence to the end of “Model Evaluation and Performance” subsection under “Results” to reference these results:

To understand how much our model improves over known risk factors, we fitted a linear regression model to the training (ADNI) set using ordinary least squares (OLS), in which the dependent variable was AD diagnosis, and independent variables were left hippocampus volume, right hippocampus volume, age, estimated total intracranial volume, and sex. In supplementary table S7 it is possible to see that our model is significantly better.

Reviewer #3, comment #7

The authors state in the abstract, introduction and discussion that structural imaging studies on preclinical AD have been mainly done on genetic forms of AD, whereas it is unclear if structural changes can be detected in sporadic AD. This is not correct as the literature on structural imaging changes in the preclinical phase of sporadic AD is vast and goes back of 10-15 years: please see, as an example, Tondelli et al 2012, den Heijer et al. 2006, Smith CD et al 2007 to quote some of the earliest studies.

Our response #3.7

We are very grateful to the reviewer for pointing out these important studies. We have amended the relevant part of the introduction to the following:

Predicting disease with such certainty before symptom onset is not possible in sporadic forms of dementia, so an alternative strategy is needed to identify an at-risk population using disease biomarkers to find people with early stages of neuropathology who are at high risk of developing cognitive impairment in the future. Small studies of ageing cohorts capturing people who have converted to Alzheimer's disease have identified group-level structural changes in the medial temporal lobe detectable prior to diagnosis [15][2][13][9]. Whether identifying such changes are sufficient to identify individuals at risk of dementia is unclear. Identifying such a high-risk group would be suitable for prevention studies or early disease-modifying treatment trials [6].

and in the discussion:

The earliest studies to identify prediagnostic structural brain changes of Alzheimer's disease found group level differences in medial temporal structures [2][15], but with low sensitivity (78%) and specificity (75%) [13]. A recent study using ADNI data in a 32 people who progressed to MCI and 8 to AD confirmed group-level structural changes in medial temporal structures that were detectable 10 years prior to onset.

Reviewer #3, comment #8

The statement "the role and timing of b-amyloid PET abnormalities remain uncertain in the detection of presymptomatic Alzheimer's disease" in the discussion is arguable and misleading. Please note that markers of amyloid deposition are now considered biomarkers of preclinical AD (see Sperling et al 2011)

Our response #3.8

In research, beta-amyloid is widely used as a biomarker. However, in clinical practice the high false positive rate in the elderly population make it difficult to interpret. We amended that sentence to the following one to make our point clearer:

so whilst PET may be helpful for risk stratification, the role and timing of β -amyloid PET abnormalities remain uncertain in the detection of presymptomatic Alzheimer's disease in a clinical setting

Reviewer #3, comment #9

I strongly suggest refraining from using the expression "AD positive participants" to indicate subjects who resulted to have a positive AD score: please do keep referring to the "score". The distinction is crucial, and relates to my major observation above, i.e. the model identifies subjects who might possibly have preclinical AD on the basis of structural imaging, but this is not proven.

Our response #3.9

We agree with the reviewer on this point, and are grateful for helping us focus on the score. We amended the manuscript accordingly where the expression was being used.

Reviewer #3, comment #10

In the discussion it is stated "key cognitive domains of AD including memory and fluid intelligence": this is not correct (see above)

Our response #3.10

We believe that our clarifications in our response # 3.2 should have also addressed this comment.

References

- [1] C. Carbone, E. Bardi, M. G. Corni, L. Fiondella, S. Salemm, G. Vinceti, M. Tondelli, M. A. Molinari, A. Chiari, and G. Zamboni. Fluid intelligence and its neural correlates in early onset mild cognitive impairment. *Alzheimer's & Dementia*, 17(S6), dec 2021.
- [2] T. den Heijer, M. I. Geerlings, F. E. Hoebeek, A. Hofman, P. J. Koudstaal, and M. M. B. Breteler. Use of hippocampal and amygdalar volumes on magnetic resonance imaging to predict dementia in cognitively intact elderly people. 63(1):57–62. ISSN 0003-990X. doi: 10.1001/archpsyc.63.1.57.
- [3] C. Fawns-Ritchie and I. J. Deary. Reliability and validity of the UK biobank cognitive tests. *PLOS ONE*, 15(4):e0231627, apr 2020.
- [4] Y. Gal. *Uncertainty in Deep Learning*. PhD thesis, University of Cambridge, 2016.
- [5] K. D. Harrington, C. Dang, Y. Y. Lim, D. Ames, S. M. Laws, R. H. Pietrzak, S. Rainey-Smith, J. Robertson, C. C. Rowe, O. Salvado, V. L. Villemagne, C. L. Masters, and P. Maruff. The effect of preclinical alzheimer's disease on age-related changes in intelligence in cognitively normal older adults. *Intelligence*, 70:22–29, sep 2018.
- [6] B. Imtiaz, A.-M. Tolppanen, M. Kivipelto, and H. Soininen. Future directions in Alzheimer's disease from risk factors to prevention. *Biochemical Pharmacology*, 88(4):661–670, Apr. 2014. ISSN 0006-2952. doi: 10.1016/j.bcp.2014.01.003.
- [7] D. P. Kingma and J. Ba. Adam: A method for stochastic optimization. In *3rd International Conference on Learning Representations, ICLR 2015, Conference Track Proceedings*, 2015.
- [8] J. K. Kruschke. Rejecting or Accepting Parameter Values in Bayesian Estimation. *Advances in Methods and Practices in Psychological Science*, 1(2):270–280, June 2018. ISSN 2515-2459. doi: 10.1177/2515245918771304. URL <https://doi.org/10.1177/2515245918771304>. Publisher: SAGE Publications Inc.
- [9] S. Kulason, E. Xu, D. J. Tward, A. Bakker, M. Albert, L. Younes, and M. I. Miller. Entorhinal and transentorhinal atrophy in preclinical alzheimer's disease. 14. ISSN 1662-453X. URL <https://www.frontiersin.org/articles/10.3389/fnins.2020.00804>.
- [10] A. Paszke, S. Gross, F. Massa, A. Lerer, J. Bradbury, G. Chanan, T. Killeen, Z. Lin, N. Gimelshein, L. Antiga, A. Desmaison, A. Kopf, E. Yang, Z. DeVito, M. Raison, A. Tejani, S. Chilamkurthy, B. Steiner, L. Fang, J. Bai, and S. Chintala. Pytorch: An imperative style, high-performance deep learning library. In H. Wallach, H. Larochelle, A. Beygelzimer, F. d' Alché-Buc, E. Fox, and R. Garnett, editors, *Advances in Neural Information Processing Systems 32*, pages 8024–8035. Curran Associates, Inc., 2019.
- [11] F. Pedregosa, G. Varoquaux, A. Gramfort, V. Michel, B. Thirion, O. Grisel, M. Blondel, P. Prettenhofer, R. Weiss, V. Dubourg, J. Vanderplas, A. Passos, D. Cournapeau, M. Brucher, M. Perrot, and E. Duchesnay. Scikit-learn: Machine learning in Python. *Journal of Machine Learning Research*, 12: 2825–2830, 2011.
- [12] S. Seabold and J. Perktold. statsmodels: Econometric and statistical modeling with python. In *9th Python in Science Conference*, 2010.

- [13] C. D. Smith, H. Chebrolu, D. R. Wekstein, F. A. Schmitt, G. A. Jicha, G. Cooper, and W. R. Markesbery. Brain structural alterations before mild cognitive impairment. 68(16):1268–1273. ISSN 0028-3878, 1526-632X. doi: 10.1212/01.wnl.0000259542.54830.34. URL <https://n.neurology.org/content/68/16/1268>. Publisher: Wolters Kluwer Health, Inc. on behalf of the American Academy of Neurology Section: Articles.
- [14] N. Swaddiwudhipong, D. J. Whiteside, F. H. Hezemans, D. Street, J. B. Rowe, and T. Rittman. Pre-diagnostic cognitive and functional impairment in multiple sporadic neurodegenerative diseases. Preprint at <https://doi.org/10.1101/2022.04.05.22273468>, apr 2022.
- [15] M. Tondelli, G. K. Wilcock, P. Nichelli, C. A. De Jager, M. Jenkinson, and G. Zamboni. Structural MRI changes detectable up to ten years before clinical alzheimer’s disease. 33(4):825.e25–825.e36. ISSN 0197-4580. doi: 10.1016/j.neurobiolaging.2011.05.018. URL <http://www.sciencedirect.com/science/article/pii/S0197458011002016>.
- [16] Y. Yao, A. Vehtari, D. Simpson, and A. Gelman. Using stacking to average bayesian predictive distributions (with discussion). 13(3):917–1007. ISSN 1936-0975, 1931-6690. doi: 10.1214/17-BA1091. URL <https://projecteuclid.org/journals/bayesian-analysis/volume-13/issue-3/Using-Stacking-to-Average-Bayesian-Predictive-Distributions-with-Discussion/10.1214/17-BA1091.full>. Publisher: International Society for Bayesian Analysis.

REVIEWERS' COMMENTS:

Reviewer #1 (Remarks to the Author):

I have reviewed both the response to reviewers document and the revised manuscript. In my opinion, the authors have adequately addressed both my critiques of the original manuscript and those of the other two reviewers. I appreciate their careful attention to the comments and suggestions. The revised manuscript is clearer and much improved, and I have no remaining concerns.

Reviewer #2 (Remarks to the Author):

I am happy with all responses and the modifications from the authors. Good work!

Reviewer #3 (Remarks to the Author):

The Authors have addressed all my comments.